# Expectation violations signal goals in novel human communication

Tatia Buidze [1] ✉, Tobias Sommer [1], Ke Zhao[2,3], Xiaolan Fu [2,3] & Jan Gläscher [1] ✉

Communication, often grounded in shared expectations, faces challenges when a Sender and Receiver lack a common linguistic background. Our study explores how people instinctively turn to the fundamental principles of the physical world to overcome such barriers. Specifically, through an experimental game in which Senders convey messages via trajectories, we investigate how they develop novel strategies without relying on common linguistic cues. We build a computational model based on the principle of expectancy violations and a set of common universal priors derived from movement kinetics. The model replicates participant-designed messages with high accuracy and shows how its core variable—surprise—predicts the Receiver's physiological and neuronal responses in brain areas processing expectation violations. This work highlights the adaptability of human communication, showing how surprise can be a powerful tool in forming new communicative strategies without relying on common language.

Effective communication relies on the ability of the Sender to design messages that clearly convey intentions, and the ability of the Receiver to accurately interpret these messages. In verbal communication, this is achieved as speakers and listeners track the statistical properties of language to optimize communication, often selecting the most predictable outcomes to facilitate understanding[1]. The Rational Speech Act model, for instance, posits that speakers optimize utterances to minimize the listener's surprise while maximizing the likelihood of accurate interpretation by the listener[2]. Similarly, the Information Density Hypothesis suggests that speakers manipulate syntactic structures to maintain a consistent flow of information, by aligning their linguistic choices with the statistical regularities of speech[3].

However, this predictability, seen in natural language processing, is not always maintained in communication. In many instances, communication takes on unconventional forms to achieve a specific goal. For example, in signaling behavior where the Sender needs to strategically direct the attention of the Receiver, it has been demonstrated that inefficiencies can signal particular intentions[4], objects, even those not intrinsically communicative like chairs or ropes, can convey social meanings through their contextual arrangement[5], and non-verbal cues

can embody complex concepts like time[6]. These examples, while diverse in communicative strategies, highlight a common theme: the strategic deviation from expected communicative patterns to achieve greater clarity and intentionality in communication. These deviations are particularly effective because they contrast sharply with anticipated behaviors, thereby capturing attention and signaling intentions more distinctly.

Here, we propose a unified framework of expectation violation as a mechanism for the signaling function of human communication. This approach entails two main mechanisms. First, when faced with situations lacking a common language, humans instinctively revert to more basic, universal properties of the physical world (e.g., movement direction or velocity) to form a shared understanding. It is not about discovering new universal principles; rather, it is about repurposing universally understood properties to build a new common communicative system through repeated social interactions. Second, once these common grounds are established, they serve as the prior expectations from which Senders can intentionally deviate. These deviations are used to create surprise, which directs the Receiver's attention to essential information and transforms into communicative acts.

[1]Institute of Systems Neuroscience, University Medical Center Eppendorf, Hamburg University, Hamburg, Germany. [2]State Key Laboratory of Brain and Cognitive Science, Institute of Psychology, Chinese Academy of Sciences, Beijing, China. [3]University of Chinese Academy of Sciences, Beijing, China. ✉ e-mail: tatiabuidze@gmail.com; glaescher@uke.de

To substantiate this proposed framework, we developed a computational model known as the Surprise model and tested its effectiveness within the context of the Tacit Communication Game (TCG). TCG is an experimental task designed to elicit novel communicative messages through spatial trajectories on a square game board[7–10]. Each player, known as the Sender and the Receiver, has distinct goals, but only the Sender is aware of both positions. The Sender's objective is to create a trajectory (the message) that begins at her starting point and leads toward her goal location, while simultaneously indicating the Receiver's goal location (see Fig. 1).

Given that TCG focuses on spatial movement and trajectories, in the Surprise model, we developed two types of priors: one based on movement kinetics, such as the expectation that a moving object will continue its path unless acted upon by an external force[11], and the other based on the Sender's goal orientation. These priors create the core of a new common language, where trajectories along a straight line signify a direction and embody a shared expectation. Deviations from these paths thus transform into communicative acts. The Surprise model utilizes these priors to construct messages that maximize information-theoretic surprise, which quantifies the psychological state of surprise at the Receiver's goal state. By leveraging these deviations from expected paths, communicative acts become signals with meaning through social interactions.

Here, we show that the Surprise model, which uses expectation violations as a core signaling mechanism, effectively replicates participants' behavioral data across two independent samples by generating messages that closely resemble those created by human Senders. Furthermore, we demonstrate that model-derived step-by-step surprise—resulting from intentional deviations from established priors—significantly influences both pupillary dilation responses (PDR) and EEG measurements in all Receivers. Specifically, the model's surprise values positively correlate with neural activity in fronto-central EEG sensors above the anterior cingulate cortex (ACC), a brain region typically associated with detecting expectation violations[12,13]. Additionally, we observe a positive correlation between surprise values and changes in pupillary dilation, which are typically linked to detecting unexpected environmental changes[14–16]. These findings indicate that Receivers adeptly detect unexpected shifts in communication, as evidenced by their PDR and neural responses. Overall, our results support the framework that expectation violations serve as a fundamental mechanism for signaling in human communication, effectively linking information-theoretic surprise to measurable physiological and neural responses.

## Results

We explored how humans form novel communication systems without a shared language, and how deviations from expected patterns can be used to convey information in the TCG. This game allowed us to estimate the effectiveness of the Surprise Model in replicating human behavior and the physiological and neural correlates of surprise during non-verbal communication. Our goal was to demonstrate that humans can use basic principles of movement kinetics to create new communication norms, and that these norms can be systematically studied and modeled.

We first provide a brief overview of the TCG. Next, we introduce the computational models, followed by a model-free analysis of behavioral data. Building upon this, we proceed to validate the computational models and explore physiological observations related to non-verbal communication within the TCG framework.

### Experimental task

The TCG is a cooperative, two-player game where the Sender and Receiver must establish non-verbal communication. In the game, the Sender knows both players' destinations and creates a path (the message) to guide the Receiver. The Receiver then uses this message to find his target location. The game unfolds over five primary steps (see Fig. 2a). Initially, the complete goal configuration, including the starting location, the Sender's goal, and the Receiver's goal, is fully visible to the Sender but not to the Receiver. Specifically, the Sender has access to all three elements of the configuration, whereas the Receiver is only presented with the starting location and the Sender's goal, leaving the Receiver's own goal undisclosed. Next, the Sender moves her token

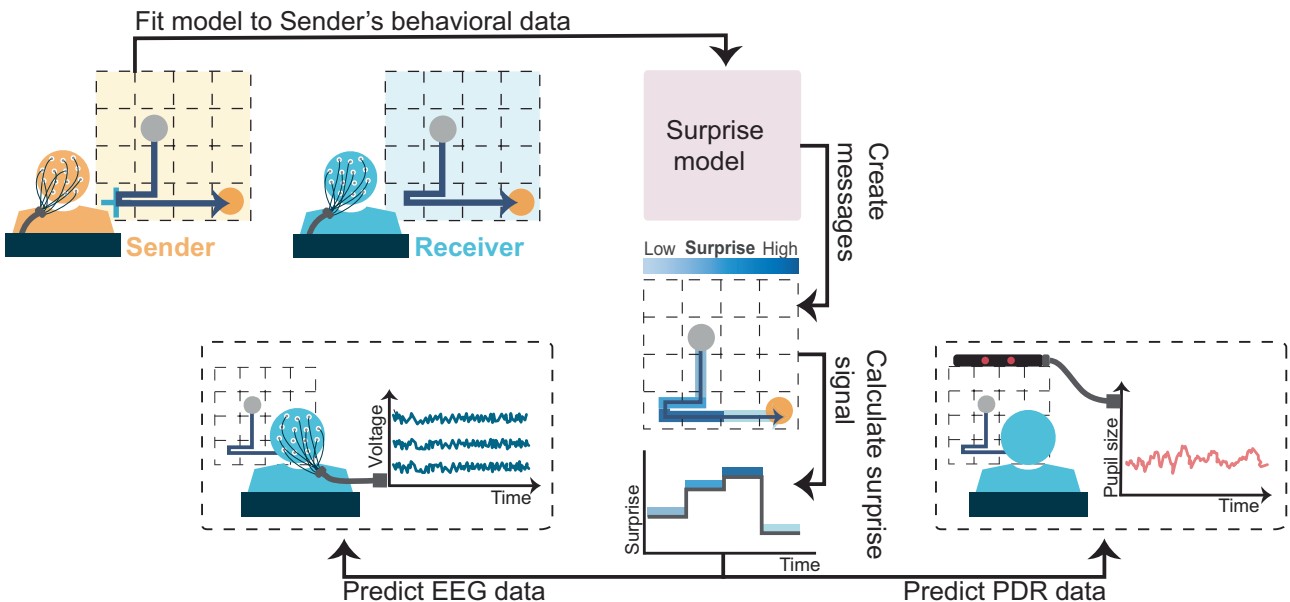

**Fig. 1 | Overview of experimental procedure and analysis.** This figure depicts an experimental framework utilizing the Surprise model within the TCG. The experiment begins by recording behavioral data from both Senders and Receivers as they engage in TCG. This recorded data is then used to fit the Surprise model to estimate the individual parameters. With these parameters, the model generates communicative messages, each accompanied by message steps and their corresponding surprise values. These values are subsequently analyzed to assess their influence on the Receiver's physiological responses, specifically through EEG and PDR measurements.

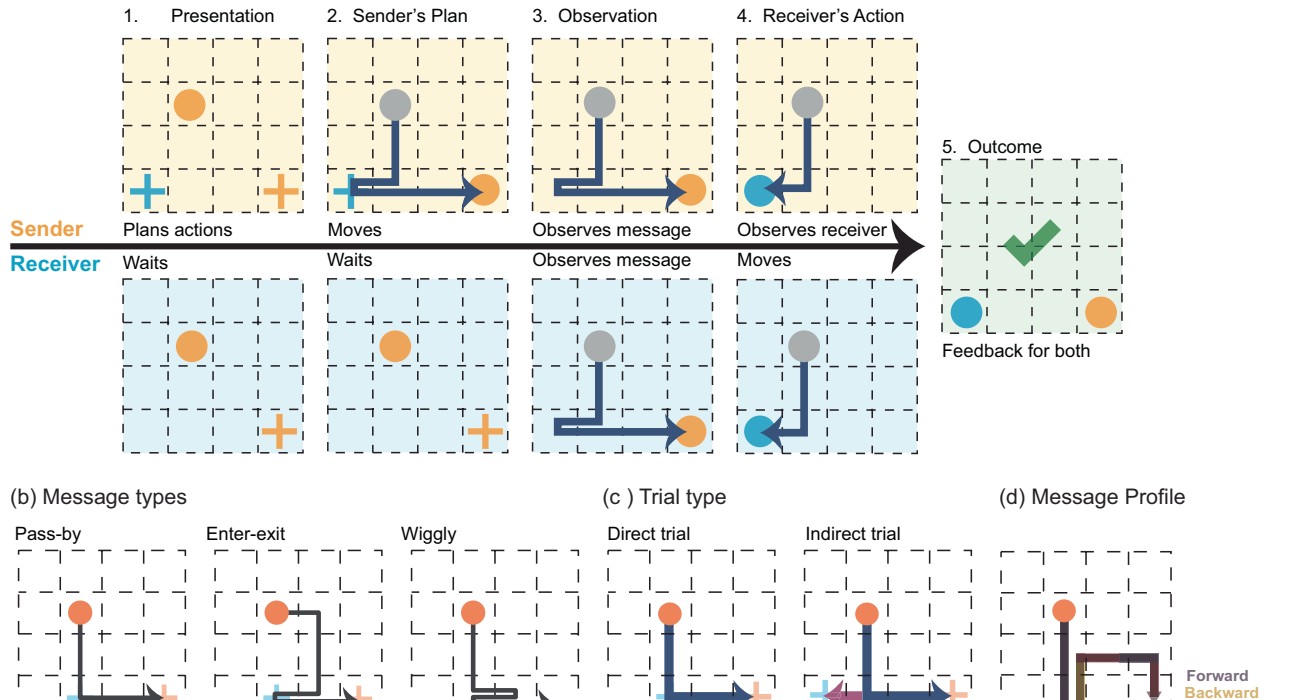

**(a)** Tacit Communication Game Trial

**Fig. 2 | Trial steps and goal configurations in the modified TCG. a.1** Presentation: our version of TCG begins with players being presented with the relevant goal configuration. The presented goal configuration for the Sender always includes three elements: the starting location (orange circle), the Receiver's goal location (blue cross), and the Sender's own goal location (orange cross). For the Receiver, it comprises the starting location (blue circle) and the Sender's goal location (orange circle). Both the Sender and the Receiver start at the same location on the grid. **a.2** Sender's plan: the Sender designs a message by moving her token first to the Receiver's goal and eventually to her goal. **a.3** Observation: the Receiver and the Sender both examine the path crafted by the Sender. This step was incorporated to facilitate the simultaneous recording of EEG data from both participants under uniform visual stimulus conditions and to equalize the step length, preventing a "pause" strategy to indicate the Receiver's goal location (see supplementary text and Fig. S1 for more details). **a.4** Receiver's action: the Receiver moves towards the inferred goal. **a.5** Outcome: success is displayed if the Receiver's choice is correct, indicated by a green checkmark. **b** Message types: three message types were created by the participants. **c** Trial types: an illustration of two different trial types. **d** Message profile: an illustration of a message profile featuring three characteristics. The direction of movement is always defined relative to the previous movement. The initial movement originating from the starting location does not have a predefined direction. However, subsequent movements are oriented based on the preceding movement.

toward her destination while encoding the Receiver's goal location in her movements, creating a 'message' to inform the Receiver of his goal. Following this, both participants observe the trajectory created by Sender. The Receiver then interprets this trajectory and moves his token where he thinks his goal state is. A successful trial occurs when the Receiver reaches the correct location, granting both players an equal reward. The reward is quantified as ten points reduced by the number of steps the Sender used to transmit the message.

The TCG consists of two different types of goal configurations: 'Direct' and 'Indirect' (Fig. 2c). Direct goal configurations can be communicated by the shortest path from the starting location to the Sender's goal location, while indirect goal configurations require a deviation from the direct path. Direct Trials are inherently more difficult for the Receiver because his goal state does not stand out from the direct path to the Sender's goal state. In contrast, when the Receiver's goal is not on the direct path (i.e., an Indirect Trial) the Sender must step sideways to enter the Receiver's goal state. This difference also prompts the creation of different Message Types (Fig. 2b).

We collected behavioral data from two distinct samples (subsequently referred to as dataset 1 (D1) and dataset 2 (D2) consisting of 58 and 62 participants, respectively). Eye-tracking and EEG hyperscanning data were exclusively obtained in D2 experiments. All pairs played 120 trials of the TCG. The two datasets differed in the proportion of direct and indirect trials: D1 had 66% indirect and 34% direct trials, while D2 had 76% indirect and 24% direct trials.

## Information-theoretic surprise as the communication signal

In this section, we provide a narrative overview of the Surprise model. The mathematical details can be found in the "Methods" section. The Surprise model consists of three distinct components: (i) the Movement model, detailing the step-by-step priors over all available actions, (ii) the State model, providing a prior guiding message creation toward the Sender's goal, and (iii) the Reward, providing an orientation to the Receiver's goal and an urgency toward shorter messages. We tested the necessity of these components in a formal model comparison, in which we compared degenerate versions (Movement and State models) against the full Surprise model.

The Movement model defines priors for all possible movements in each state. These priors are based on a fundamental idea from movement kinetics: without an external force, a moving object is expected to continue its motion in the same direction where it came from. Any deviation from this direction has a lower prior probability and is therefore more surprising. This leads to the rule illustrated in Fig. 3a, where $p$ (forward) $> p$ (left) $= p$ (right) $> p$ (backward). The calculation of these subject-specific priors is governed by two estimable parameters, $\gamma$—the probability of a backward movement, and $\lambda$, an odds-ratio parameter that describes the ratio of priors for forward movement and leftward or rightward movement. Movement priors are dynamic and can change with every step along the message path.

The State model defines State priors, which provide a general sense of the goal orientation for the Sender and decrease in concentric

## Computational Models of Message Creation

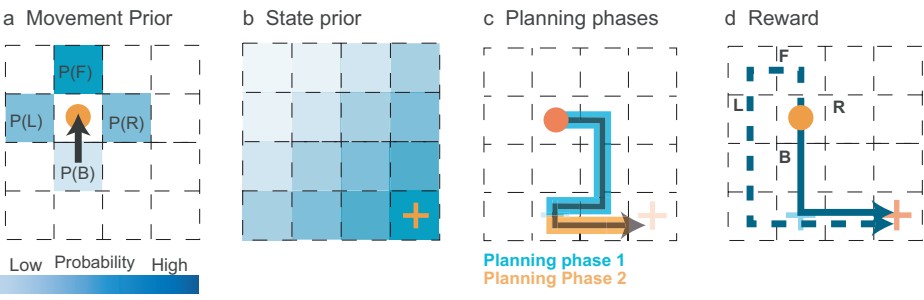

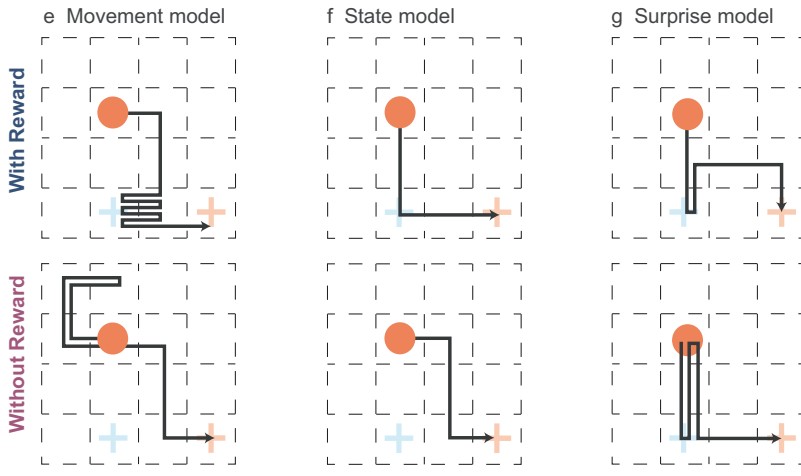

**Fig. 3 | Computational models of message creation. a** Movement prior assigns the highest probability to forward movement, then left/right, and least to backward. **b** State prior conveys the Sender's goal by reducing probabilities at each state with increased distance. **c** Planning phases: during Phase 1, the Sender communicates the Receiver's goal; in Phase 2, the Sender proceeds to her own goal. **d** Reward: Illustrates higher rewards for backward movements, facilitating shorter paths to goals. **e**–**g** Model components: simulations with each model component alone reveal that the Movement model (**e**) emphasizes the Receiver's goal with surprising events, while the State model (**f**) focuses solely on the efficient Sender's goal achievement. The Surprise model (**g**) combines the efficiencies of the Sender's goal achievement (State priors) with the emphasis on signaling for the Receiver's goals (Movement priors). Comparing messages with reward information (first row) to those without (second row) demonstrates that incorporating reward minimizes cost and efficiently signals the Receiver's goal. These messages also match human messages most closely.

circles from her goal according to an inverse power rule (Fig. 3b). Thus, moving along an increasing State prior will move the Sender straight toward her goal. The State prior is static for the duration of the trial, but changes with different goal configurations. Its shape is determined by a single slope $\alpha$ parameter (see "Methods" for details).

In the Surprise model, the Sender selects the step-by-step actions by intentionally violating expectations that are defined by both types of priors: Movement and State priors. This novel policy of action selection by surprise lies at the core of the model and is the basis of its success in modeling message design in the TCG. The process of creating a message involves two distinct phases: (i) in the first phase of the message, Movement and State priors govern action selection from the start location to the Receiver's goal state. (ii) In the second phase of the message from the Receiver's goal state to the Sender's goal state, the Sender aims to reach her goal state as quickly as possible (Fig. 3c). Here, in the second phase, the principle of selecting a surprising action is abandoned in favor of a policy that is solely governed by the State prior.

Rewards for each potential next action are calculated as the discounted cost (i.e., number of steps) to reach the Sender's goal state via the shortest path through the Receiver's goal state. Rewards provide an urgency to reach the Sender's goal quickly to retain the maximum points from the initial endowment. The discount rate is estimated with a single free parameter $\varepsilon$ (see "Methods" for details). For example, Fig. 3d showcases a scenario where the action of moving backward (B)

is linked to a greater reward due to its ability to reach goal locations via the shortest path (depicted by the blue solid line), while other actions (F, L, R) incur higher costs (indicated by a dashed line for action F).

The combination of the surprise from the Surprise model and the rewards yields an expected value over all available next actions given a specific planning horizon (see "Methods" for details). These expected values are filtered through a softmax function (parameterized with a choice stochasticity parameter $\tau$) to render the final action probabilities for the next action.

Figure 3e–g illustrates the effect of each model component on message planning in an intuitive way. Simulations with only the Movement model (Fig. 3e, top) reveal that the Sender will perform many zigzagging moves, because of the high surprise of the backward action. In contrast, the pure State model (Fig. 3f, top) creates a direct path message without surprising moves. The combination of both models (Surprise model) (Fig. 3g, top) generates messages comparable to the human-created Enter-Exit messages. The second row illustrates the effect of the different model components without the reward. The pure Movement and State models (Fig. 3e, f, bottom) generate similar message patterns, but fail to communicate the Receiver's goal. The Surprise model (Fig. 3g, bottom) generates 2-step wiggles, but without the reward component, this model does not advance to the final Sender's goal efficiently. In Summary, by combining Movement and State priors along with the surprise values and discounted future rewards, the Sender can effectively communicate the Receiver's goal

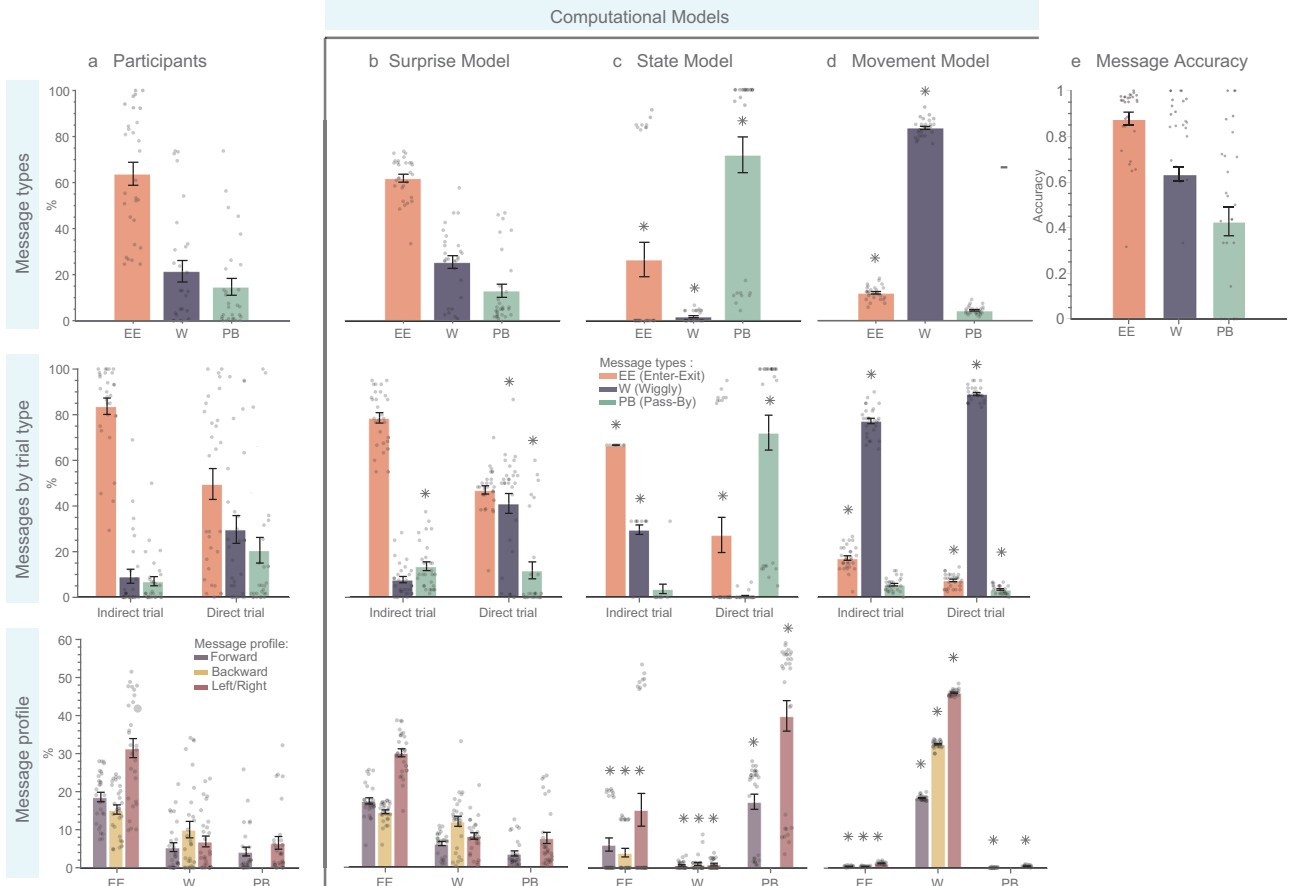

**Fig. 4 | Comparing participant-generated and model-simulated messages for dataset 1. a** Participants: participants' behavioral data categorized by message types, trial type, and message profile. **b** Surprise model simulated data after model fitting exhibits similarity to participants' data (Bayes factor analysis was used to compare models. $BF_{01} > 1$, indicating evidence for no difference). **c** State model and **d** Movement model simulated data reveal differences from participant's data (asterisks indicate $BF_{01} < 1$, suggesting evidence against similarity). **e** Message accuracy: accuracy as the Receiver's success in finding his goal state. The Enter-Exit message has the highest accuracy rate, followed by Wiggly and Pass-By messages. All error bars represent means ± SEM. Sample size ($n = 29$) represents independent biological replicates, with each data point corresponding to a unique participant. Figure created using Matlab R2023a.

location. Once this objective is accomplished, the State priors then guide the Sender towards taking the shortest path to reach her own goal location.

**Communicative strategies in TCG: a model-free analysis**
Using our experimental data, we developed different indices to describe the behavior of the Sender to gain a more detailed understanding of the cognitive processes involved in planning new communicative messages (For detailed results on dataset 1 (D1) and dataset 2 (D2), see Figs. 4a and 5a, respectively). These indices are also used in posterior predictive checks to test whether the fitted model can generate simulated data that closely resembles the observed experimental data.

We identified three most common and distinct message types that Senders used to communicate their intentions to Receivers. These message types are illustrated in Fig. 2b (see "Methods" for details). Enter-Exit message was predominantly used by participants (D1: M = 63.82%, SE = 5.00%; D2: M = 54.95%, SE = 6.6%), and involves Senders momentarily diverging from the direct route to the Receiver's goal before proceeding to their own goal, noted for its high success rates of 87% in D1 and 88% in D2. Wiggly message ranks as the second most common (D1: M = 21.47%, SE = 4.67%; D2: M = 26.65%, SE = 5.91%), and features Senders navigating on the direct path, shifting between the Receiver's goal and the next state to create a zigzag pattern, achieving accuracy rates of 63% in D1 and 76% in D2. Pass-By message is

the least frequently employed but still notable (D1: M = 14.70%, SE = 3.66%; D2: M = 18.41%, SE = 4.95%). It involves Senders passing through the Receiver's goal area on their way to their own goal and has the lowest observed accuracy rates of 42% in D1 and 29% in D2. Individual participant message profiles and strategy changes over the course of the experiment are detailed in the Supplementary Material.

Our findings indicate that participants utilized all message types in both trial contexts, but with different frequencies. Notably, during the Direct Trial, there was a marked increase in the use of Wiggly and Pass-By messages, suggesting a preference to remain on the direct route during these trials. Conversely, in the Indirect trial, the most used message type was the Enter-Exit message (see the second rows of Figs. 4a and 5a for D1 and D2 results, respectively). A comprehensive breakdown of these findings is provided in Supplementary Tables S1b and S2b.

The message profile details the number of movements in each direction, relative to the last movement into the current state. For instance, the Enter-Exit message profile depicted in Fig. 2d includes movements Forward, Left, Right, and Backward. Each message type exhibited a specific profile. Enter-Exit messages are characterized by frequent left/right movements, followed by forward and backward movements. Wiggly messages have a higher frequency of backward movements (due to the zigzag pattern), followed by left/right and then forward movements. Pass-By messages differ from the other message types by exclusively relying on left/right and forward movements,

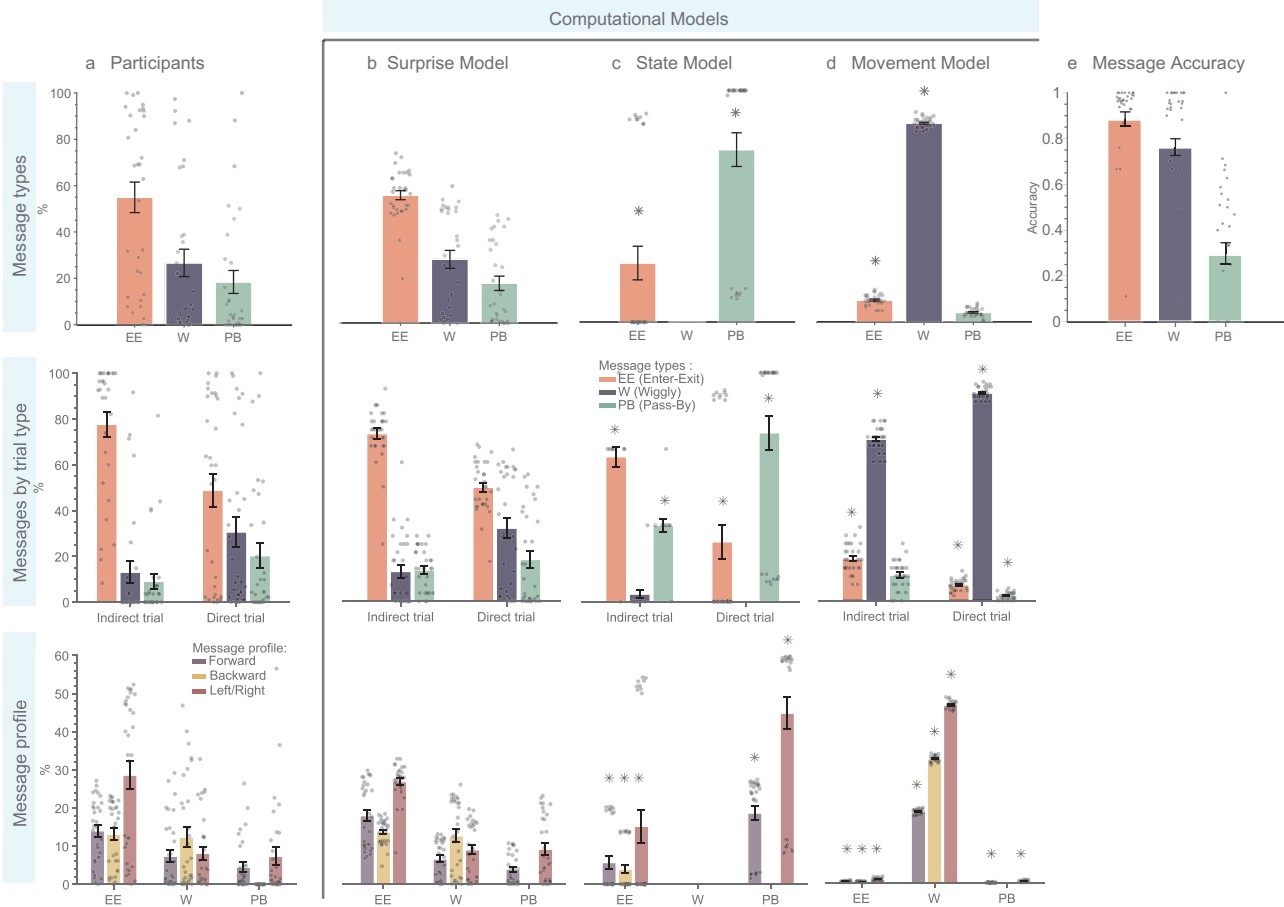

**Fig. 5 | Comparing participant-generated and model-simulated messages for the dataset 2. a** Participants: participants' behavioral data categorized by Message types, Messages elicited in different types of trials, and Message profile. **b** Surprise model simulated data after model fitting exhibits similarity to participants' data (Bayes factor analysis was used to compare models. $BF_{01} > 1$, indicating evidence for no difference). **c** State model and **d** Movement model simulated data reveal differences from participant's data (asterisks indicate $BF_{01} < 1$, suggesting evidence against similarity). **e** Message accuracy: accuracy is calculated as the Receiver's success in finding his goal state. Enter-Exit messages have the highest accuracy rate, followed by Wiggly and Pass-By messages. All error bars represent means ± SEM. Sample size ($n = 31$) represents independent biological replicates, with each data point corresponding to a unique participant. The figure was created using Matlab R2023a.

without incorporating any backward movement. Exact frequencies can be found in the Supplementary Material (Table S1c for dataset 1 and Table S2c for dataset 2).

In summary, this analysis highlights how Senders adapt their communication strategies within the TCG, opting for different message types depending on the trial type. The analysis highlights the cognitive flexibility required to navigate the game's constraints, revealing that different strategies significantly impact communication success. These insights into message usage and effectiveness are crucial for understanding the underlying cognitive processes and refining cognitive models of nonverbal communication that we will present in the next section.

**Surprise model accurately explains Sender's message design**
To determine which model variant most accurately reflects the observed behavioral data, we fitted the free parameters for each model by minimizing the negative log-likelihood of the model for the collected trial-by-trial data across subjects. The resulting model likelihoods, model frequencies[17], and protected exceedance probabilities (PEP)[18] are outlined in Table 1 for dataset 1 and Table 2 for dataset 2. Here, log-likelihood serves as a cost function for model fitting, with higher values indicating a more precise fit. Model frequency reflects the extent of a model's suitability across different subjects, and PEP evaluates the probability of a model being the most appropriate, while also considering the potential that no model may ideally fit the data.

The Surprise model fitted the observed data best. Its corresponding model frequencies—0.80 for Dataset 1 and 0.78 for Dataset 2—suggest strong consistency in explaining the behavior across the majority of participants. This is further substantiated by the PEP; High PEP, close to 1 for both datasets, signals a strong statistical confidence that the Surprise model is the best model among the three variants tested. This measure of confidence also considers the null hypothesis that no model in the model space is likely across the population. In contrast, the Movement and State models alone exhibit significantly lower PEP, which translates to a lower chance of these models being the best-fitting explanation for the datasets. This convergence of likelihood, frequency, and PEP distinctly positions the Surprise model as the most effective model for capturing the nuances of the observed behavioral data.

**The surprise model predicts participants' behavior across different samples**
Next, our objective was to evaluate the ability of the Surprise model to accurately replicate participant behaviors across different samples. We fitted the Surprise model to the initial sample using hierarchical Bayesian estimation[17] and used the estimated group parameter values to simulate the message types and profiles for the observed sample. To quantitatively assess the congruence between actual and predicted frequencies, we employed Bayes factor analysis[19]. The majority of

**Table 1 | Comparative evaluation of model performances for dataset 1: likelihoods, frequencies, PEP (protected exceedance probabilities), and posterior group parameters**

| Model | Log likelihood | Model frequency | PEP | $\tau$ | $\varepsilon$ | $\lambda$ | $\gamma$ | $\alpha$ |
|---|---|---|---|---|---|---|---|---|
| Surprise model | −60.06 | 0.80 | 1.00 | 6.93 | 0.39 | 0.10 | 2.14 | 1.84 |
| State model | −70.38 | 0.16 | 0.00 | 2.90 | 0.39 | – | – | 1.11 |
| Movement model | −96.57 | 0.05 | 0.00 | 1.92 | 0.62 | 0.15 | 1.50 | – |

Model: refers to the three tested computational models: surprise, state, and movement models. Log likelihood: represents the log-transformed probability of the data given the model, where higher values indicate a better fit of the model to the observed data. Model frequency: reflects the proportion of subjects for which a given model is most likely the best fit, with values closer to 1 indicating higher generalizability across the sample population. PEP: is the probability that a model is more likely than any other model, while also considering the possibility that no single model is a good fit for the data population. Posterior group parameters ($\tau$, $\varepsilon$, $\lambda$, $\gamma$, $\alpha$): estimated group values for parameters: $\varepsilon$ (discount rate for planning horizon), $\lambda$ (backward movement), $\gamma$ (forward and left/right movement), $\alpha$ (slope of State prior), and $\tau$ (choice stochasticity).

**Table 2 | Comparative evaluation of model performances for dataset 2: likelihoods, frequencies, PEP (protected exceedance probabilities), and posterior group parameters**

| Model | Log likelihood | Model frequency | PEP | $\tau$ | $\varepsilon$ | $\lambda$ | $\gamma$ | $\alpha$ |
|---|---|---|---|---|---|---|---|---|
| Surprise model | −60.46 | 0.78 | 0.99 | 6.80 | 0.31 | 0.11 | 2.03 | 1.83 |
| State model | −69.48 | 0.22 | 0.00 | 2.77 | 0.45 | – | – | 1.15 |
| Movement model | −94.26 | 0.00 | 0.00 | 2.50 | 0.70 | 0.15 | 1.98 | – |

Model: refers to the three tested computational models: Surprise, State, and Movement models. Log likelihood: this represents the log-transformed probability of the data given the model, where higher values indicate a better fit of the model to the observed data. Model frequency: reflects the proportion of subjects for which a given model is most likely the best fit, with values closer to 1 indicating higher generalizability across the sample population. PEP: is the probability that a model is more likely than any other model, while also considering the possibility that no single model is a good fit for the data population. Posterior group parameters ($\tau$, $\varepsilon$, $\lambda$, $\gamma$, $\alpha$): estimated group values for parameters: $\varepsilon$ (discount rate for planning horizon), $\lambda$ (backward movement), $\gamma$ (forward and left/right movement), $\alpha$ (slope of State prior), and $\tau$ (choice stochasticity).

Bayes factors obtained underscored a notable similarity across samples, as evidenced by $BF_{01}$ values ranging from 1.2 to 11.6 (for details, see Table S3). This analysis demonstrates that the Surprise model is not merely the most suitable fit for the behavioral data, but also emphasizes its capability to accurately mirror behaviors across varied datasets.

## Surprise model outperforms simpler model variants

Here we compare message generation by the Surprise model and its degenerate versions (i.e., State, and Movement models) with the messages generated by the participants. We employed a Bayesian approach[19] to assess the support for two hypotheses: the null hypothesis ($H_0$), which suggests no difference between the messages generated by each model and the empirical data, and the alternative hypothesis ($H_1$), which posits substantial differences between the model and the empirical data. We quantified the strength of this evidence for or against these hypotheses using a Bayes Factor ($BF_{01}$) for each behavioral index. Deviations of the model generated from the behavioral data ($BF_{01} < 1$) are marked with asterisks above the model-generated indices in Figs. 4 and 5 (for details, see Table S1 and S2).

When comparing the means of human-created message types with those of the Surprise model, the resulting $BF_{01}$ (D1: $BF_{01} = 10.20$ (Enter-Exit), $BF_{01} = 6.48$ (Wiggly), $BF_{01} = 10.36$ (Pass-By); D2: $BF_{01} = 11.12$ (Enter-Exit), $BF_{01} = 10.95$ (Wiggly), $BF_{01} = 10.95$ (Pass-By)) strongly support the null hypothesis that human-created message types are similar to the ones generated by the Surprise model. We conducted a similar analysis for message profiles, revealing moderate to strong evidence supporting $H_0$. This is indicated by $BF_{01}$ values for various profile elements, ranging from 1.31 to 10.42 (details in Tables S1c and S2c for D1 and D2, respectively). These findings suggest that messages generated by the Surprise model are largely similar to human message profiles, though there are notable exceptions regarding Wiggly and Pass-By messages in different trial contexts of dataset D1. Unlike the Surprise model, comparisons between human-created messages and those from the State and Movement models favor the alternative hypothesis, showing that these models differ from human-generated messages and do not accurately capture human message generation. For more detailed information, please see Table S2b.

In summary, formal model comparison demonstrated that the Surprise model was most effective in capturing the nuances in participants' behavior. It showed high consistency across datasets and was able to accurately replicate behaviors, emphasizing its robustness and superiority over the simpler models in understanding non-verbal communication strategies.

## PDR correlates with model-derived surprise

Previous studies have demonstrated a close link between pupillary dilation responses (PDR) and violations of expectations (i.e., a surprise)[14]. We therefore hypothesized that the step-by-step surprise values calculated by the Surprise model would correlate significantly with the Receiver's PDR while observing the Sender's messages. To test this, we devised a model-informed analysis, in which we used the step-by-step surprise values from the Surprise model as a predictor for the participant's step-by-step PDR and used a mixed-effects model as implemented in the lmer package in R to test this correlation[20]. In the model, we incorporated a fixed effect for the task condition—a continuous variable representing surprise values. We also included random intercepts and slopes for each participant. The mean PDR change of message steps served as the predicted variable. The residuals of the model met the assumptions of normality and homoscedasticity.

The results revealed that the baseline-corrected PDR (w.r.t. to a pre-stimulus baseline of -200ms to onset) exhibited a significant effect for surprise ($F (1,20) = 60.20$), $p < 0.001$, eta = 0.75, 95% CI [0.55, 1.00], two-tailed test). Specifically, the analysis demonstrated a significant relationship between continuous surprise values and PDR, as visualized in Fig. 6a. The fixed effect estimate for surprise (0.02) indicates that for each unit increase in surprise, the baseline-corrected PDR increases by approximately 0.02 units. This positive linear relationship shows that greater surprises induce more pronounced physiological responses, as measured by pupillary dilation.

In summary, we investigated how surprise, predicted by the Surprise model, correlates with pupillary dilation responses (PDR). The analysis shows a significant correlation between surprise and PDR. This

a  Predicted PDR

b  Expected Surprise Levels during Observation

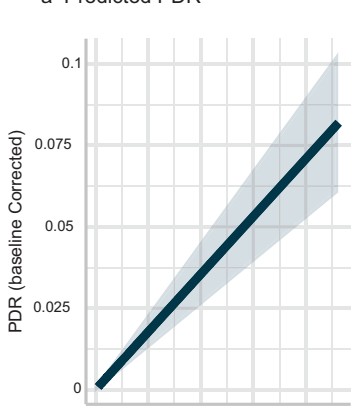

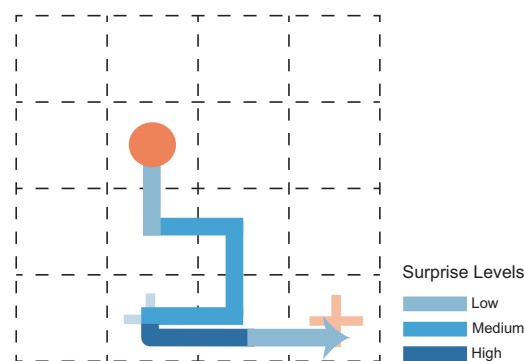

**Fig. 6 | Model-based analysis of PDR data. a** Predicted PDR: the graph shows the predicted baseline-corrected PDR as a function of model-derived surprise values. The solid line represents the mean predicted PDR, while the shaded area indicates the 95% confidence interval around the mean. The relationship demonstrates a significant linear increase in PDR with increasing surprise values. **b** Surprise levels during observation: as the message steps are observed by the Receiver, varying levels of surprise associated with different steps are expected. Certain steps are more surprising than others.

suggests that surprise effectively triggers measurable physiological responses, underscoring its impact in non-verbal communication settings.

### Neural encoding of surprise in TCG

We hypothesized that if Receivers identify their goal location based on the deviation from their expectations, then we should observe a neuronal encoding of surprise in brain regions associated with error detection and conflict resolution[21,22].

We used the step-by-step surprise values from the Surprise model again as a predictor for EEG power in a model-informed EEG analysis. Specifically, we followed a trial-by-trial model-based approach[23–26] in which for each Receiver, we performed a regression at each electrode and time point within 0–1200 ms of each message step. Subsequently, we analyzed the regression weights by testing them against 0 across subjects for all electrodes and time points.

First, we identified clusters of electrodes and time points that showed significant sensitivity to the predictor surprise irrespective of message type, after correcting for the multiple comparisons (The cluster formation threshold was set at $p = 0.005$ for $t$-tests)[27]. Specifically, we observed two distinct time-space clusters correlating significantly with surprise across electrodes and time. The first cluster displayed a positive effect at the frontal electrodes between 400 ms and 800 ms of each message step. This cluster is visually depicted in Fig. 7a, with yellow markers ($p < 0.05$ cluster corrected). The second cluster showed a positive effect at the frontal-central electrodes between 800 ms and 1200 ms. This cluster is highlighted in Fig. 7a with pink markers, also denoting significance ($p < 0.05$ cluster corrected). This analysis suggests that information-theoretic surprise is encoded in the neural activity levels in the brain regions identified by the significant clusters.

To further explore the neural origin of these sensor-level findings, we conducted a source localization analysis using individual participants' structural MRI images and electrode coordinates. The resulting source activity indicated activation in regions close to the supplementary motor area (SMA) and the dorsal anterior cingulate cortex (dACC), as shown in Fig. 7b.

We also observed that different message types (Enter-Exit, Wiggly, and Pass-By) are associated with different levels of surprise. Is this difference also reflected in the neural activity in these areas? To answer this question, we repeated the above sensor-level analysis for each message type separately. Initially, we computed the average of the EEG data for each message type, followed by applying the same regression analysis at every time point and electrode as previously described. Subsequently, we examined the average regression weights from the clusters identified as significant in the earlier analysis (frontal and central clusters). The bar plots in Fig. 7c next to the regression weight time course show the average surprise levels associated with each message type as calculated by the Surprise model. We observed that message types characterized by high surprises, like Enter-Exit and Wiggly, exhibited a stronger correlation with EEG signals from fronto-central electrodes compared to less surprising messages such as Pass-By.

In summary, an analysis of neural encoding of surprise in EEG data unveiled two prominent time-space clusters over frontal and fronto-central electrodes, potentially linked to surprise processing. High-surprise message types, such as Enter-Exit and Wiggly, showed stronger correlations with EEG signals from fronto-central electrodes compared to less surprising messages like Pass-By, suggesting distinct neural activity patterns linked to varying levels of surprise in communication. Additionally, source localization analysis indicated activation in the SMA and dACC regions, suggesting these areas as the neuronal origin of the observed sensor-level cluster.

### Discussion

In this study, we explored the role of surprise as a communicative signal in the absence of a shared language among participants by employing the TCG as an experimental platform. The Surprise model capitalizes on the information-theoretic surprise as the signal most relevant for selecting the next possible action. A surprise can only unfold based on prior expectations, which the model calculates by integrating dynamic Movement priors, rooted in principles of movement kinetics, and static State priors encoding the general goal orientation. Additionally, a reward component pushes towards creating shorter, more efficient messages that also pass through the Receiver's goal. Model simulations (Fig. 3 e-g), as well as model comparison with degenerate versions of the model (i.e., State model and Movement model, Table 1), revealed that each component is necessary to generate messages similar to those created by human participants in two independent data sets. The interplay of these model components suggests that message planning in the TCG is a complex cognitive operation, wherein human players intricately balance different subgoals (represented by the model components) to communicate efficiently.

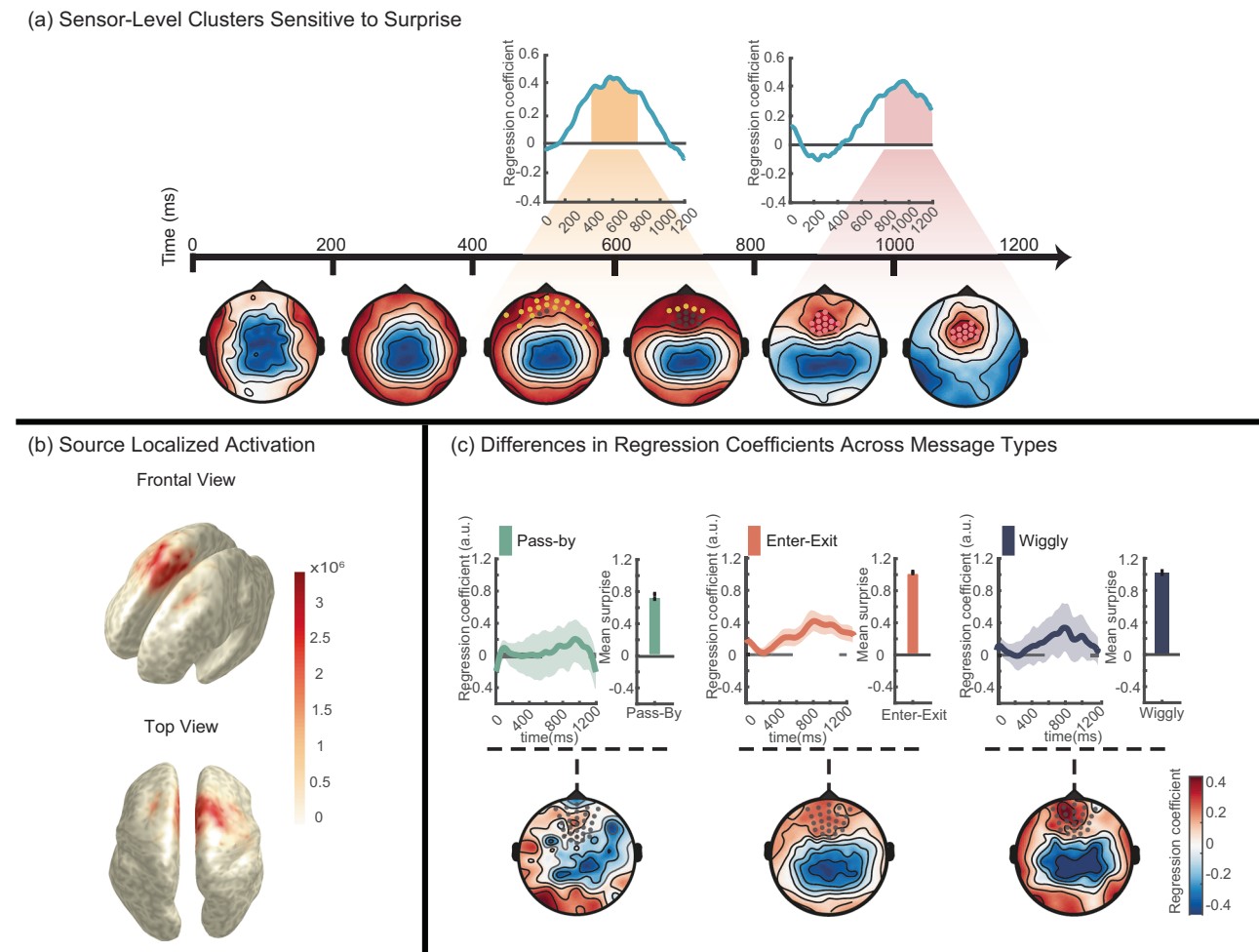

**Fig. 7 | Model-based analysis of EEG data. a** Two distinct electrode clusters sensitive to surprise: the first cluster showed positive responses in frontal electrodes (represented by yellow dots on the topoplot at 400–800 ms), while the second occurred in frontal-central electrodes (illustrated by pink dots on the topoplot at 800–1200 ms). The time axis is the epoch of one message step during the observation phase of each message (see Fig. 2a) which always lasts for 1200 ms. **b** Source localized activation: source localization analysis revealed activation in the supplementary motor area (SMA) and the dorsal anterior cingulate cortex (dACC) when participants process surprising communicative signals. **c** Average regression weights for different message types: regression weights were averaged over all participants for each message type and across Central-frontal electrodes (gray

dots). Line plots display the mean regression coefficients over time, with shaded areas representing the standard error of the mean (SEM) around the mean. Bar plots display the mean surprise per message type as calculated by the model. Error bars on the bar plots represent means ± SEM. Message types such as Enter-Exit and Wiggly, which have higher surprise values, correlate more strongly with EEG signals from Central-frontal electrodes compared to less surprising types like Pass-By. The sample size ($n$ = 31) represents independent biological replicates, with each data point corresponding to a unique participant. This figure was generated in Matlab R2023a using the FieldTrip toolbox[59], which is licensed under CC BY-SA 4.0 (https://creativecommons.org/licenses/by-sa/4.0/).

The key insight of this study is that a framework of expectation violation can serve as a mechanism for signaling goal orientation in human communication. Specifically, when humans are confronted with social situations in which they cannot rely on shared cultural, linguistic, or even gestural cues, they revert to a more fundamental level of physical laws that govern our environment and are universally understood. These form the basis for commonly understood priors on which a new communicative system can unfold through repeated social interactions. Using this canvas of priors, humans can then employ an equally fundamental mechanism—expectancy violations—to communicate important information and guide Receivers' attention.

This approach contrasts with traditional communication models that rely on predictability and shared linguistic cues[2,3]. These are more naturalistic scenarios where reliance on universal knowledge is unnecessary because the presence of shared language enables predictability to serve as the primary driver of language comprehension. For example, listeners and readers often use contextual information to

predict upcoming linguistic input[1], and predicting this input facilitates a faster understanding of the message[28]. This predictability of a word in the context of a sentence is described by quantities from information theory, namely the KL divergence (or Shannon surprise)[29]. This theoretical framework partially aligns with our Surprise model, which also uses Shannon's surprise to create messages that are informative for the Receiver in finding their goal, provided they can decode the surprise by evaluating steps with respect to the priors, constructed from universal principles of movement kinetics. However, unlike traditional models that focus on minimizing surprise to enhance comprehension, our framework maximizes surprise. This approach is particularly crucial in the absence of a shared language base, where maximizing surprise helps convey information more effectively.

While predictability is the primary mechanism in classic communication models, there are situations where expectancy violations can also play a crucial role in effective communication. For example, classic work on implicature shows that being unnecessarily verbose can convey additional meaning because the verbosity violates the

expectation of conciseness[30]. Similarly, in prosody, unexpected variations in pitch, tone, or rhythm can enhance the emotional impact and engagement of verbal communication[31]. More recent research has explored the adaptive benefits of expectancy violations. Repeated exposure to syntactic anomalies, such as garden path sentences, helps individuals adapt and process language more effectively over time[32]. Moreover, strategically introducing unexpected lexical or syntactic complexity can also enhance the persuasive impact of messages, demonstrating how deliberate expectancy violations improve communication effectiveness, even in contexts requiring high levels of clarity and engagement[33].

Similar principles have been observed in other referential communicative situations as well. Participants interpret communicative actions through an expectation for efficient communication, often perceiving inefficient actions as intentional when they are unexpected[4]. This supports our observation that unexpected actions in the TCG are often the most communicatively effective. Additionally, research indicates that when interpreting color adjectives, people prioritize context-specific informativity over general world knowledge[34]. In experiments, participants were more likely to choose a less typical yellow object, such as a chair, over a more common one, like a banana, when the color adjective played a crucial role in identification. This finding suggests that, in language processing, context-specific expectations can override general assumptions based on world knowledge.

Comparing these two mechanisms of communication—reducing surprise through predictability and increasing surprise by deviating from expectation—reveals that human communication fundamentally involves managing expectations. While using surprise is more typical in novel communication without a shared language, it can also be employed in verbal communication. Introducing unexpected elements in verbal interactions can enhance engagement, memorability, and effectiveness. Examples include jokes, which rely on unexpected punchlines to elicit laughter and engage the audience[35], and prosody, which uses variations in pitch, tone, and rhythm to maintain interest and enhance comprehension[36]. These two strategies of managing expectations demonstrate the adaptability of human communication across various contexts.

While our Surprise Model maximizes unexpectedness to enhance communication without a shared language, traditional frameworks including reinforcement learning[37], predictive coding and Bayesian inference[38], and active inference[39,40] handle surprise and prediction errors differently. These models use prediction errors, deviations between expectations and outcomes, as teaching signals to refine future expectations, aiming to select actions that minimize these errors for accurate prediction and long-term rewards. In contrast, our study shows that in communication, Senders use surprise to disrupt prior consistent perceptions and convey critical information. This action selection by surprise contrasts with traditional action selection by value, producing messages that mimic human design in scenarios like the TCG. Thus, our model highlights a unique policy of maximizing surprise, and adaptable based on task demands.

An alternative to expectation management in communication is employing the Theory of Mind (ToM). ToM models form beliefs about a partner's behavior and choose actions accordingly[41–43]. In the TCG context, a ToM model requires defining a belief distribution over possible states and exhaustively searching for each goal configuration, demanding extensive memory and contradicting the brain's preference for efficiency. The Belief-based Model (BBM)[44] is an example of such a model that includes ToM-like mentalization, but it operates on the entire message, thus precluding step-by-step analyses. We qualitatively compare the BBM to the Surprise model by simulating message selection for all experimental goal configurations, but because of the impossibility of calculating a subject-specific likelihood, the BBM was not included in the formal model comparison. These simulated messages from the BBM and a Bayes factor analysis showed that the model fails to capture human message generation, suggesting ToM is not the primary mechanism in TCG communication. Detailed information about the BBM structure and simulation outcomes are available in the supplementary material.

We observed supporting evidence for the Surprise model in the analysis of the physiological and neural data. Traditionally, variations in pupil diameter have been indicators of arousal[45–47] and cognitive effort[48]. More recently, these changes have been reinterpreted to provide insights into factors such as surprise, salience, decision biases, and other dynamics that influence information processing[14,49,50]. A significant portion of these physiological responses can be attributed to the activation of norepinephrine (NE)-containing neurons in the locus coeruleus (LC) of the brainstem[51,52]. In examining the Receivers' physiological reactions while processing messages, we found a direct correlation between PDRs and model-derived surprise signals. This indicates that Senders, when intentionally communicating with surprise, have a significant impact on the physiological states of Receivers.

Sensor-level analysis of the neural encoding of surprise in the EEG data revealed two prominent time-space clusters over frontal and fronto-central electrodes. Source localization analysis identified a dipole, the potential neuronal origin of these clusters, in the region encompassing both the Supplementary Motor Area (SMA) and the dorsal anterior cingulate cortex (dACC). The first cluster, appearing around 400 ms after the onset of the communication step, may reflect the Receiver mentally simulating the Sender's intended motor actions while observing the message, potentially involving SMA activity, which is typically engaged in motor planning[53]. The second cluster, observed later (800 ms after the onset), may be associated with the detection of deviations from priors, involving the (dorsal) anterior cingulate cortex (ACC), a region known for error detection and the integration of diverse information to guide decisions aligned with goals and reward representations[12,13]. The ACC is also thought to play a crucial role in linking actions to outcomes and adjusting perceptions based on new evidence[54,55]. Recent studies suggest its involvement in discerning others' intentions and detecting unexpected outcomes[56,57]. Our findings align with this perspective, and we interpret the second cluster as potentially reflecting ACC activity related to the Receiver's processing of surprise as intentionally designed by the Sender[22]. However, we recognize that these interpretations remain exploratory, and further studies are needed to confirm the functional specificity of these findings.

The potential implications of these findings extend across multiple domains, emphasizing the importance of harnessing expectancy violations to improve communication in areas such as negotiation, human-machine interfaces, and AI technologies. In negotiation, utilizing strategic signaling and managing expectations can capture attention and shift dynamics. Beyond negotiation, integrating elements of surprise into human-machine interfaces can make interactions more engaging and intuitive. Unexpected yet relevant system responses can capture user attention and enhance learning and retention, leading to more user-friendly and effective interfaces. Integrating this concept with AI technologies could potentially make AI more human-like and creative. Generative AIs trained on extensive text typically analyze sets of possible words based on contextual information to ensure clarity and coherence by selecting the most probable words. Even though these selection algorithms occasionally introduce some variability through top-k sampling and temperature scaling, they only ensure moderate variability in word selection. Applying the principles of our Surprise model would lead to the selection of greatly unexpected words, thus ensuring even greater variability in the selection process. This adaptation would be an approach, that potentially diverges from predictable paths to enhance creativity and engagement. Such an application could enhance AI's capabilities in comedy, storytelling, or generative art, creating more engaging and unique scenarios.

While communication through surprise is crucial in the TCG and has broad implications, it is not the exclusive method for novel environments or general communication. The TCG provides a controlled setting for studying non-verbal communication and expectation violations but does not fully capture the complexities of real-world communication, which often involves a mix of verbal and non-verbal cues and relies on social and emotional contexts. To increase the ecological validity of these results in future research, we could start by translating the game from the non-verbal to the verbal domain. Another modification could involve allowing participants to choose between cooperative and competitive behaviors to explore different social contexts. Finally, we could alter the game design to observe if the intention to communicate can naturally emerge and how surprise plays a role in this process. For instance, participants could learn to communicate while playing the game without explicit instructions to do so.

In summary, the Surprise model demonstrates the effectiveness of incorporating surprise in communication without a shared language, a phenomenon explored through the TCG. Not only did the Surprise model exhibit a notable alignment with human Sender behaviors, but it also validates its impact through both physiological and neural markers, affirming the crucial role of surprise in novel communication.

## Methods

### Participants

This study complies with ethical regulations approved by the General Medical Council of the City of Hamburg (PV7114). All participants provided written informed consent prior to the study and were financially reimbursed for their participation. Each participant received a base payment of 12 euros for every hour of participation in the experiment. In addition to the base payment, participants were eligible for a bonus (Mean bonus: 4 euros) based on their performance.

Dataset 1 involved 58 individuals (mean age: M = 25, SD = 4.4), consisting of 29 pairs of Senders and Receivers, while dataset 2 included 62 participants (mean age: M = 26, SD = 4.1), comprising 31 pairs of Senders and Receivers. Study participants were recruited through university job boards. The Exclusion criterion was a prior history of neurological or psychiatric diseases. All the participants were naive to the task and had been instructed with standardized written instructions.

### Experimental procedure

Prior to the experiment, the participants were placed in the same room in front of separate screens to play the game after they had been assigned roles and given detailed standardized instructions. Instructions informed them that success in the game depended on both players reaching their respective goal locations. Specifically, the Sender was aware of both players' destinations and needed to guide the Receiver to his undisclosed goal location. Additionally, both players were told each movement cost a score, and the bonus at the end of the game would be calculated based on the sum of the scores from all the trials. We clarified that movement on the grid was restricted to horizontal or vertical directions. Importantly, there were no practice rounds offered, allowing us to observe the whole spectrum of behavior from the moment participants were exposed to the game rules until the end. The instructions deliberately avoided mentioning any specific strategies like managing expectations or deviations, to prevent biasing participants towards any predetermined strategies. Our goal was to capture a wide range of natural communication behaviors as players adapted to the game's requirements. The TCG game consisted of 120 trials. In every trial, players had to solve different goal configuration problems. A goal configuration consists of three elements: (1) the starting location in one of four middle states of the 4 × 4 game board (same for both players), (2) the Sender's goal location, and (3) the Receiver's goal location.

At the beginning of every trial, the Sender was presented with a complete goal configuration, while the Receiver could only observe the starting location and the Sender's goal location (Fig. 2a). After seeing the goal configuration and planning the message, the Sender pressed a button to indicate that she was ready for movement. She then had 5 s to move her token on the grid. Once the Sender had reached her goal state, the entire message was shown to both players with the same step duration of 1.2 s. After the message replay had finished, the Receiver's token appeared at the starting location, and he started moving after pushing the start button. Finally, both players received feedback regarding their accuracy as a team.

To differentiate between different goal configurations, we focused on the location of the Receiver's goal relative to the direct path as the key trial characteristic. This characteristic was used to define two distinct types of goal configuration, namely Direct, and Indirect, as depicted in Fig. 2c. The goal configuration, where the Receiver's goal location can be effectively conveyed by the shortest path from the starting location to the Sender's goal location (Direct path) is referred to as the Direct configuration. Conversely, the goal configuration in which the Receiver's goal location can only be communicated by diverting from the Direct path is referred to as the Indirect configuration. In the experiment, Dataset 1 comprised 66% of trials with a Direct Goal Configuration and 34% with an Indirect Goal Configuration. In contrast, Dataset 2 exhibited a 76% Direct Goal Configuration to 24% Indirect Goal Configuration trial ratio.

### Model-free behavioral indices

We defined different indices to characterize the Sender's behavior in a more fine-grained way, providing insights into the cognitive processes behind creating new messages. These indices cover the message's general category (Message type) and its specific characteristics (Message profile).

We've identified three Sender-generated message types: Enter-Exit, Wiggly, and Pass-By (as shown in Fig. 2b). The Enter-Exit message involves the Sender moving towards the goal and then reversing direction, essentially entering, and then exiting the area near the goal. The Wiggly message, while similar to Enter-Exit in its backward movement, is distinguished by its multiple entries and exits, creating a series of back-and-forth movements between two locations. In contrast, the Pass-By message is characterized by a straightforward movement from the starting point to the goal, without any backward or repeated movements.

The message profile is a detailed characteristic of each message and includes information on the number of movements in each direction, specifically, forward, backward, left, and right. An example profile of the Enter-Exit message is shown in Fig. 2d, which consists of 4 forward, 2 left and right, and 1 backward movement.

### Analysis of pupillary responses

**Data acquisition and preprocessing.** Pupillary dilation data of the Receiver were collected using the screen-mounted eye-tracker iView Red-m (Sensorimotor Instrument (SMI), Teltow, Germany) with 120 Hz sampling frequency, while the Receiver was observing the Sender's message (Fig. 2a), displayed on the screen with the resolution of 1920 × 1200 pixels. The eye-tracker was positioned 60 cm away from the participant's face. A calibration was done before each game with the iViewX software. To avoid pupil dilation caused by the sunlight, the windows in the testing room were covered with blackout curtains with a light-blocking effect. We used the same brightness and screen settings for all the experiments.

To identify and remove the noise and invalid data segments from the data, we followed the Guidelines recommended by Kret et al.[58] (https://github.com/ElioS-S/pupil-size) to prepare and filter raw data.

After identifying and interpolating invalid samples, we selected the relevant 1.2 s epochs during the replay phase of the trial for further

analysis. Pupillary dilation response (PDR) data were downsampled by a factor of 10, thus rendering each step as a time series of 120 data points.

**Mixed-model analysis for pupillary data.** We investigated the effect of surprise values generated by the Surprise model on the Receiver's PDR, using a mixed-model analysis implemented in the lme4 package in R version 3.6.1, with the parameters estimated through maximum likelihood estimation[20].

In the model, we included a fixed effect for surprise values, and a random intercept and slope for each participant:

$$PDR \sim surprise + (1 + surprise \mid sub) \quad (1)$$

To test the significance of the fixed effects, we used the likelihood ratio test (LRT) implemented in the lme4 package in R[20].

## EEG data analysis

**Data preprocessing and cleaning.** EEG signals were continuously recorded from both players with a 128-channel Active 2 system from BioSemi (Amsterdam, Netherlands) (0.1–100 Hz band pass; 2048 sampling rate). In the current study, we only analyzed the Receiver's EEG data during the replay section of a trial (Fig. 2a). We divided the continuous EEG time series into segments from 0 to 1200 ms based on the onset of each message step. After visually inspecting the data, we identified and interpolated an average of 6 bad channels out of 128 total channels. Additionally, we rejected an average of 40 bad epochs out of 550 total epochs. Blinks were removed by using Independent Component Analysis (average number of removed components: 5). The electrodes were re-referenced to the average common reference across the channels. These preprocessing steps were carried out using the EEGLAB software version 2022.1.

**Model-informed regression analysis.** We used a model-informed approach to analyze the EEG data[23–26]. The approach entails a linear regression analysis of the EEG data of all steps of a message onto the Surprise model-derived surprise values for each message step at all-time points in the epoch and across all electrodes.

For this regression analysis, data were bandpass filtered from 0.5 Hz to 40 Hz, and downsampled to 256 Hz. Baseline correction was conducted by subtracting the mean activity between 0 and +100 after movement onset because there was no gap between different movement steps (and hence no time for the traditional pre-stimulus baseline) and because inspection of the ERPs did not show any significant deflection in the first 100 ms of each message step. We performed the above-mentioned regression analysis for each participant at each electrode and time point within 1200 ms epoch following movement onset. The predictors were the model-derived surprise values (described below) for each message step and a trial-specific intercept. This procedure creates a time series of regression coefficients, which were averaged across trials and then tested against 0 across participants to determine the time point at which the model-derived surprise significantly predicted the epoched EEG data.

The multiple comparison problem was addressed by using a cluster-based permutation test, as implemented in the field trip software package[59]. The cluster formation threshold was set at a significance level of $p = 0.005$ for the $t$-test.

**Neural source localization analysis.** The neural sources associated with the observed EEG signals were identified through a source localization analysis, which involved several key steps:

(1) MRI acquisition and preprocessing: high-resolution T1-weighted structural MRI images were obtained for each participant (3D MPRAGE sequence, TR 2300 ms, TE 2.89 ms, flip angle 9°, 1-mm slices, field of view 256 × 192; 240 slices). Alignment of the MRI coordinate system to the CTF coordinate system facilitated accurate integration with EEG

electrode coordinates. Coregistration of MRI images with EEG electrode positions was performed interactively, ensuring anatomical information from the MRI matched the spatial configuration of the EEG sensors. The MRI images were resliced for consistent orientation and voxel size, improving accuracy in segmentation and source localization.

(2) Segmentation and head model creation: automated algorithms segmented the MRI images into brain, skull, and scalp tissues, providing necessary anatomical details for constructing head models (part of the field trip package). A boundary element model (BEM) was created from the segmented tissues, accurately representing the conductive properties of each participant's head. EEG electrode positions that were generated from 3D images recorded from each participant's head while still wearing the EEG caps were aligned to the scalp mesh through interactive procedures, ensuring accurate correspondence between electrode positions and anatomical structures.

(3) Source reconstruction analysis: the lead field matrix was calculated using the head model and aligned electrode positions, describing how each potential source contributes to the observed EEG signals. Time-locked analysis of the Receiver's EEG data for specified time windows generated averaged data representing neural activity during each message step. Source analysis was conducted using linearly constrained minimum variance (LCMV) beamforming, which computes neural sources by maximizing signal from a given location while minimizing contributions from other locations. Source data for each participant were averaged to create a grand average source activation map, representing common neural sources across all participants. The grand average source data were normalized to MNI standard space and interpolated onto MRI images, allowing visualization of neural sources in a standardized anatomical space.

## Surprise model

The core concept behind the Surprise model for the TCG is to communicate the Receiver's goal location by defying the expectations enforced by probabilistic priors about the next likely step and the general direction of the Sender's goal location. Violating these expectations communicates to the Receiver that he must pay attention to this part of the message. The model quantifies these expectation violations by calculating the information-theoretic surprise for each available action and then selecting the action with the highest surprise. Thus, a critical part of the model is the definition of the prior probabilities. The model incorporates two types of priors—movement priors and State priors.

**Movement priors.** The Movement model assigns probabilities to all available actions, based on the concept of movement kinetics, that an object in motion without an external force is likely to continue moving in a straight line. Deviations from this trajectory are less likely and, therefore, more surprising. This notion implies a dynamic realignment of the Movement priors after each step. For instance, if the model takes a turn, the action probabilities will shift towards the new straight direction. Generally, the movement probabilities follow the ranking order of

$$p(f) > p(l) = p(r) > p(b) \quad (2)$$

where $p(f)$ is the probability of going forward, $p(l)$ is the probability of going left, $p(r)$ is the probability of going right, and $p(b)$ is the probability of going backward. These four probabilities are parameterized using two estimable parameters, namely, $\lambda$ and $\gamma$. They are expressed in the model as follows:

$$\lambda = p(b) \quad (3)$$

$$\gamma = \frac{p(f)}{p(l) + p(r)} \quad (4)$$

$\lambda$ is restricted to be small (e.g., 0.001–0.3), whereas $\gamma$ is restricted to a range between 2–10. For instance, if $\gamma = 2$, then $p(f)$ is twice as likely as the combined probability of a turn $(p(l) + p(r))$ and hence, $p(f) = 2/3$ and $p(l) = p(r) = 1/6$.

More generally, the individual probabilities for all four movements can be calculated from these two parameters in the following way:

$$p(b) = \lambda \tag{5}$$

$$p(f) = \frac{\gamma}{\gamma + 1}(1 - \lambda) \tag{6}$$

$$p(l) = p(r) = 0.5\left(1 - \frac{\gamma}{\gamma + 1}\right)(1 - \lambda) \tag{7}$$

Note that the Movement priors remain constant for all steps in all trials, but the orientation (forward, left, right, backward) changes with every step the player takes.

## State priors

State priors play a crucial role in providing a general sense of the Sender's goal or destination. Specifically, the prior state probability is a function that has a value of 1 at the Sender's goal and then decreases according to an inverse power rule as the distance from the goal increases. The function is defined as

$$p(s) = \frac{1}{\alpha^d} \tag{8}$$

Where $p(s)$ is the probability of a state being at a distance $d$ from the Sender's goal location. $\alpha$ is the positive estimable slope parameter, determining the rate at which probabilities decrease with increasing distance from the Sender's goal. It remains constant for all trials, but the location of the peak of the State prior changes with every goal configuration.

## Combined movement-state priors and surprise

To render the final probabilities for each action ($p(a)$), movement and State priors are multiplicatively combined and normalized as follows.

$$p(a|s) = \frac{p(m)p(s'|s, m)}{\sum_{m=1}^{4} p(m)p(s'|s, m)} \tag{9}$$

Where probability $p(m)$ is one of the movement probabilities out of four possible movements (f, l, r, b), while $p(s'|s, m)$ is the State prior probability of state $s'$ after movement $m$. Note that this is not a softmax filtering, but rather a normalization to ensure valid probability values.

The information-theoretic (Shannon) surprise of each action is then calculated as:

$$h(a|s) = -log_2(p(a|s)) \tag{10}$$

Where $h(a|s)$ is a measure of information-theoretic surprise and shows, how much action $a$ in the state $s$ deviates from what the Receiver expected the Sender to do based on the combined movement and State priors. In contrast to other models of learning and decision-making (like Reinforcement Learning), the Surprise model uses the Shannon surprise for the calculation of the final action probabilities.

## Rewards

If the actions were only selected based on the Shannon surprise, then the final message trajectory would meander around the game board without efficiently reaching the Sender's goal (see simulations in Fig. 3e–g). The rewards (conceptualized as the cost per step in the

TCG) provide the Sender with an urge to reach his goal quickly and retain the maximum possible point from the initial endowment. Thus, in the model, we defined the rewards $r(s'|a, s)$ at the next state $s'$ that is reached by choosing action $a$ in state $s$, as the number of points that would remain from the initial ten points after taking the shortest route thereafter through the Receiver's goal to the Sender's goal. These rewards are multiplicatively combined with the Shannon surprise calculated above into an expected value:

$$ev(a|s) = h(a|s) \times r(s'|a, s) \tag{11}$$

## Two phases of message planning

Designing a message in the TCG has two components: (1) creating a trajectory to the Receiver's goal that informs the Receiver of the location. The Surprise model realizes this by using the deviations from prior expectations (Shannon surprise), and (2) afterward moving to the Sender's goal on the shortest route to retain the maximum possible amount from the initial endowment. Thus, the second planning phase no longer uses the Shannon surprise of the combined Movement-State prior for action selection, but simply moves along the direction of the State prior toward the Sender's goal. The expected value for this second planning phase is then:

$$ev(a|s) = p(s'|a, s) \times r(s'|a, s) \tag{12}$$

## Planning a message in steps

In the context of a decision-making problem like a TCG, an agent can use value iteration[60] to plan its next moves up to a certain point in the future (a planning horizon). The planning horizon determines the number of steps the agent plans. For example, if the planning horizon is 2, the agent will plan its actions from the current step until 2 steps into the future. The purpose of this planning horizon is to determine the optimal course of action that maximizes the agent's sum of expected future rewards.

In the Surprise model, this is done by generating a tree of possible actions and calculating the expected values of each possible action at each step of the planning horizon, while considering the potential future rewards that might come after. The expected value of getting to the next state in the policy tree from the current state $s_i$ using action $a_i$ is the following:

$$ev_{a_i|s_i} = h(a_i|s_i) \times \epsilon^i r(s'_i|s_i, a_i) \tag{13}$$

where $h(a_i|s_i)$ is the surprise associated with the action probability of getting to the next stage in the policy tree from the current state $s_i$ using action $a_i$ and $\epsilon^i$ is the discount factor at the planning step $i$, and $r(s'_i|s_i, a_i)$ is the reward available in the next state $s'$ that is reached from the current state $s$ via action $a$ in the planning step $i$. Once the model calculates the expected values for every state-action pair on the tree, it then sums up the expected values along each trajectory in the policy tree to render the final expected values for all immediately available actions:

$$EV(a|s) = \sum ev_{a_i|s_i} \tag{14}$$

These expected values are then filtered through a softmax function to render the final action probabilities:

$$Pr(a|s) = \frac{e^{\tau EV(a|s)}}{\sum_a e^{\tau E(a|s)}} \tag{15}$$

The softmax function is used to calculate the probability of taking a particular action $a$ in a given state $s$, based on the expected future

rewards associated with each action with an estimable temperature parameter $\tau$ that models the stochasticity of the choice.

## Parameter estimation

We used the Computational and Behavioral Modeling (CBM) toolbox (version 3, MATLAB R2021)[17], which uses a hierarchical variational Bayes approach to estimate the parameters of the Surprise model for each participant. The model is defined by five key parameters, represented as $\lambda$, $\gamma$, $a$, $\epsilon$ and $\tau$, which are detailed in Eqs. 3, 4, 8, 13, and 15, respectively. Our approach involved utilizing the Laplace approximation method, implemented in the toolbox, to effectively fit the model to our behavioral data.

## Statistical analysis of posterior predictive checks

One of the objectives of our study was to identify the computational account that most closely replicates the behavior of the human Sender in a posterior predictive check. We simulated new synthetic data using the fitted model parameters and calculated the model-free behavioral indices that were defined for the behavioral data (message frequency and message profile). We employed a Bayesian approach[19] to compare the null hypothesis ($H_0$: no differences between model-generated data and human experimental data) with the alternative hypothesis ($H_1$: substantial differences between model-generated and human experimental data) for all three model variations. A Bayesian approach allows for a quantification of evidence for both hypotheses.

To evaluate the evidence for $H_0$ vs $H_1$ for all models, we computed several Bayes factors ($BF_{01}$) by comparing the means of the behavioral indices discussed earlier with the means of the corresponding indices generated by the computational models.

A $BF_{01}$ of less than 1 favors the alternative hypothesis ($H_1$) over the null hypothesis ($H_0$), indicating that the data provides more evidence supporting the noticeable discrepancy between the model and human Sender behavior. Conversely, a $BF_{01}$ greater than 1 favors the null hypothesis ($H_0$) over the alternative hypothesis ($H_1$), indicating that the model can closely replicate the behavior of human Senders.

## Reporting summary

Further information on research design is available in the Nature Portfolio Reporting Summary linked to this article.

# Data availability

All behavioral and physiological data necessary to run the analysis are publicly available at: https://doi.org/10.5281/zenodo.14333555. The EEG data is available at: https://doi.org/10.12751/g-node.5bns43.

# Code availability

The code to run the analysis and reproduce the figures is archived and publicly available at https://doi.org/10.5281/zenodo.14333555. The repository includes a README file with detailed instructions on how to set up the environment, install necessary dependencies, and execute the code.

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

## Acknowledgements

We thank Julia Wandschura for her help with data collection and Fatemeh Hadaeghi for her valuable comments. This work is supported by a DFG/NSFC grant to J.G. and X.F. (CRC TRR 169 "Cross-modal Learning", NSFC grant 62061136001) and a DFG grant to J.G. (CRC 1528 "Cognition of Interaction").

## Author contributions

Research design—J.G. and T.B. Experimental task and data collection—T.B. Model development—J.G. Model implementation—T.B. Data analysis—T.B. Paper writing (first draft)—T.B. Paper writing (revisions)–T.B., J.G., T.S., K.Z., and X.F.

## Funding

## Competing interests

The authors declare no competing interests.
