## [Transparent Peer Review File · Nature Communications]

Expectation Violations Signal Goals in Novel Human Communication

Corresponding Author: Mx Tatia Buidze

Version 0:

Reviewer comments:

Reviewer #1

(Remarks to the Author)

SUMMARY

This is an interesting and carefully conducted study, including a behavioural experiment ran on two separate groups of human participants, which was used to collect communicative signals, pupil dilation (extracted from eye-tracking) and EEG responses. In addition, the authors built a Surprise model that predicted human responses with higher accuracy than degenerate versions of the same model (Movement and State models only). The results of the study confirm that, when deprived of a communicative system (be that language, gesture, or sign), humans can rely on universal laws of physics and expectancy violations (the key contribution of this work) to communicate a message to a receiver.

KEY RESULTS

The study explores how people instinctively turn to the fundamental principles of the physical world to overcome communication barriers. For this, they investigate how participants develop novel strategies such as movement kinetics for conveying messages in the Tacit Communication Game, without relying on common linguistic signals. They find that messages generated by the Surprise computational model—based on information-theoretic surprise—closely resemble human Sender messages and positively correlate with Receiver's physiological and neural responses. These findings enrich our understanding of surprise as a non-verbal communication tool, elucidating its computational and neural mechanisms.

SUMMARY OF STRENGTHS:

Validity: The study is carefully designed, the experimental setup reasonable, the analysis methodologically sound, and the results are both interesting and convincing.

Significance: 1) The Tacit Communication Game, where a Sender seeks to convey messages by defying the Receiver's expectations, has been used in non-verbal communication research for some time, yet the use of computational models in this area has been rather limited (as far as I know, the traditional belief-based models are unrealistic for TCG). Thus, from the modelling side, this paper pushes the field in a novel direction by proposing a computational model that detects the Sender's goal from movement patterns to explain the Receiver's understanding of the signal, plus their physiological and neural responses during non-verbal communication. This movement-based innovation is quite impressive and may generate and confirm new hypotheses about forming new communicative strategies without relying on common language. 2) From the theory side, the work nicely shows that prediction errors are more than a teaching signal improving future expectations, but also encourage efficient communication.

Data, Methodology, and Analytical approach: While I lack specific expertise in modelling the Receiver's physiological and neural responses during non-verbal communication using surprise models, my experience with agent-based modelling in communication games allows me to confirm that the data, methodology and analytical approach in this work are sound and solid. A few suggestions are made by the section below.

SUGGESTED IMPROVEMENTS

>> MAJOR

Motivation/Contribution of this paper: The significance of this paper lies in its pioneering computational framework, which contributes a theory of surprise as a potential non-verbal communication tool. While the results of this work are interesting and solid, its current presentation falls short of fully conveying its bold contributions. Therefore, there is ample room for further refinement and elaboration to underscore the innovative nature of this research.

From a theoretical perspective, this work unfortunately falls short of shedding light on real, verbal communication. The authors do not even attempt to draw any conclusions from their findings that could apply to verbal communication. I would therefore encourage the authors to engage in such discussion in order to highlight the value of their study.

I recommend the authors to contrast their notion of information-theoretic surprise (which was the driver of their results) to that of predictability in language processing (which is known to be a driver of verbal communication).

Ryskin, R., & Nieuwland, M. S. (2023). Prediction during language comprehension: what is next? *Trends in Cognitive Sciences*.

For a direct comparison of these two forces in human communication (surprise vs predictability) see also:

Rohde, H., & Rubio-Fernandez, P. (2022). Color interpretation is guided by informativity expectations, not by world knowledge about colors. *Journal of Memory and Language*.

Another point of comparison with the referential communication literature (including modelling work) is the kind of inferences that Receivers can derive from Sender's choices (e.g., an inefficient choice tends to be interpreted as an ostensive signal, the same way that participants did in this study).

Royka, A., Chen, A., Aboody, R., Huanca, T., & Jara-Ettinger, J. (2022). People infer communicative action through an expectation for efficient communication. *Nature Communications*.

Jara-Ettinger, J., & Rubio-Fernandez, P. (2021). Quantitative mental-state attributions from linguistic events. *Science Advances*.

One last interesting point of comparison between the Surprise Model reported in this manuscript and previous computational models of referential communication is the notion of Reward as a bias for shorter expressions. For a recent discussion of different cost functions in referential communication, see:

Jara-Ettinger, J., & Rubio-Fernandez, P. (2022). The social basis of referential communication: Speakers construct physical reference based on listeners' expected visual search. *Psychological Review*.

The authors are not requested to cite these works. I am just suggesting relevant literature, but they are free to review other papers and get into the same issues.

Basically, I think the Discussion section should address the following questions: what do the results of this experiment and the computational models tell us about human communication, by and large? If the principles underlying these results and models are universal, how do they transfer to linguistic communication?

>> MINOR

Discussion of AI models: The Discussion section attempts to contextualise the impact of this work within the recent advancements of large language models, proposing the consideration of a "surprise selection algorithm" that occasionally opts for less expected alternatives. While I appreciate the authors' intention, caution is warranted regarding two aspects: 1) Clarification is needed on how the action selection is generalised to word selection (e.g., referencing Shain et al. 's work). 2) Large language models do not always opt for the word with the highest probability; instead, various hyperparameters and techniques are employed to prevent the model from consistently selecting the most probable word, including top-k sampling or temperature scaling.

Shain, C., Meister, C., Pimentel, T., Cotterell, R., & Levy, R. (2024). Large-scale evidence for logarithmic effects of word predictability on reading time. *Proceedings of the National Academy of Sciences*, 121(10), e2307876121.

Writing Clarity: The writing of this work requires significant improvement in several areas 1) The introduction lacks motivation for developing movement kinematics models and fails to highlight their differences from traditional models, such as belief-based models. 2) In the results section, there is a need for more motivation before delving into the elaboration of results. It would also be beneficial to include a brief summary for each Results subsection. Additionally, many subheadings lack informativeness, such as "Predictive Accuracy of the Surprise Model across Different Sample" and "Model Fitting and Model Comparison." These headings should be grounded with specific results. 3) The work comprises multiple subparts, including experiments, modelling, and brain data analysis. It might be beneficial to include a single visualisation in the introduction that encapsulates all these subparts.

Your expertise

We are willing to defend our evaluation since it combines two different sets of skills and areas of expertise, but it's possible

we may have overlooked certain details, misunderstood key aspects, or misjudged the novelty of the work.

Reviewers:

Paula Rubio-Fernandez, PhD

Tianai Dong, PhD Candidate

Max Planck Institute for Psycholinguistics, NL

(Remarks on code availability)

We had access to the code but have not ran it on our computers.

Reviewer #2

(Remarks to the Author)

This paper investigates how humans communicate information when language and gesture are not available and the signal consists of movements in a 4x4 spatial grid. Specifically, the senders need to communicate to the receivers what spatial location they need to move to while simultaneously moving to their own (different) goal location. The authors propose that senders will draw the receiver's attention to their goal by maximizing the receiver's information-theoretic surprise which is derived from their prior beliefs about how the sender is likely to move (preferring to continue on the same path than deviate from it and to move closer toward her goal with each step). They develop a computational model which simulates the sender's messages/paths and fit its free parameters to behavioral data, demonstrating impressive posterior prediction relative to simpler models. They then show that the surprisal of a given step in the message is correlated with the receiver's pupillary response as they view the message step. Finally, they show that EEG responses collected from receivers are also sensitive to the surprise in two time windows with distinct spatial distributions.

Overall, this is a very exciting paper that should be of interest to a broad cognitive science/neuroscience audience. The writing is clear and the methods are appropriate for the questions at hand. Nonetheless, some aspects of the paper could be strengthened:

1. The title is perhaps a bit misleading. Non-verbal communication could refer to a large number of things. Often people use it to mean gesture, eye gaze, and other cues that humans use in everyday life to supplement verbal communication or when verbal communication is temporarily unavailable (e.g., when people don't want to be heard). This is quite different from the situation studied here in which communication is not simply constrained to be non-verbal, it is a novel communication system using motion in a 2D space.
2. It would be useful to add some details about the experiments. For instance, I was curious to know the exact way in participants were instructed to play the game and whether they had any practice with feedback. Did the instructions make any mention of the receiver's "expectations" or "deviations" in the path? Did participants instantly pick up on one of the three strategies or did they initially fail? It would also be interesting to know how performance changed across the trials. And, though this may be beyond the scope of the current paper, did the receiver's PDR or EEG response predict more successful communication on subsequent rounds?
3. The Belief Based Model comes out of nowhere on page 12 and doesn't even get a reference. Perhaps something got cut from an earlier section of the manuscript, but it would be helpful to explain what it is and why it is worthwhile to compare it to the Surprise model. It is actually a strength of this work that there is an alternative theoretical account which does not capture the behavioral data as well, and this should be made clear in the main body of the paper.
4. A few aspects of the statistical analysis of PDR seemed like they required some explanation. Why was surprise, a continuous variable, binned into a 3-level categorical variable? The continuous version seems more appropriate. Also, it would be appropriate to include random slopes for the (continuous) effect of surprise for each participant in the the mixed-effects models. Was this omitted due to convergence issues? (There are workarounds for that, e.g., using allFit()).
5. The EEG analysis, though a novel aspect of the study, was less compellingly presented. At a high level, the inferences that the fronto-central cluster reflected ACC activity related to prediction error and the frontal cluster reflected prefrontal activity related to cognitive control are highly speculative. They would be perhaps bolstered by attempting to tie these spatio-temporal clusters to known EEG components with well-described functional properties (e.g., P300). Otherwise, the conclusions about signal sources should be qualified. Also, it wasn't clear to me what to make of the analysis where the data were split by message type. Couldn't the strong relationship in Enter-Exit be mostly explained by the fact that there were more trials of that type?
6. In the description of the priors, the movement priors are not very intuitive. It might help to visualize how they pattern. I'm imagining maybe a raster plot with $p(f)$ on the x axis, $p(l)=p(r)$ on the y axis, and $p(b)$ represented as hue, though there might be a better way to do this.
7. I applaud the authors for making much of the data and code available on github. However, from a quick look, it didn't seem like the actual implementation of the model(s) was available in the repo, only the result of model fitting. If that's not the case, perhaps it could be more clearly documented.
8. Finally, the discussion missed the opportunity to comment on how the principles of the Surprise model would generalize to other non-grid-based domains and how we might think about scaling this up to explain more naturalistic human

communication behaviors. Moreover, it would be interesting to draw comparisons to other frameworks for communicative decisions, such as the Rational Speech Acts framework (Goodman & Frank, 2016), which have focused on situations where speakers try to minimize listeners' surprise.

(Remarks on code availability)

The code is available and largely well documented (though I did not attempt to run it myself). In my review, I pointed out one component that should be added for better reproducibility.

Reviewer #3

(Remarks to the Author)

This work addresses a very relevant topic for the study of communication and, more generally, the regulation of relationships mediated by surprise and unexpected outcomes. Through the use of neurophysiological measures (in particular EEG and pupillary dilation responses) and the application of TCG, the authors detected the presence of specific cortical networks responding to the experimental condition. The work is well articulated, handsomely solid and accurate in its discussion phase on the results. Some observations may serve to improve the current form of the paper:

- 1) the link between the theoretical introduction which largely refers to the communicative phenomena which testify to the effect of the violation of expectations (for example humour) and the present experimental setting, based on an abstract game, should be better specified, also for the purpose to understand its possible applications and implications for real communication dynamics
- 2) the reason why this task (TCG) was specifically chosen compared to other more "realistic" tasks should be better specified.
- 3) in this regard the possible limits in terms of ecological validity of the results of the present experiment should be better specified. Some suggestions for possible future expansions or modifications of the experimental setting could be added at the end of the paper
- 4) the work would benefit from a broader discussion regarding the possible implications of this type of research for application areas (for example relating to negotiation processes in real interactions).

(Remarks on code availability)

Reviewer #4

(Remarks to the Author)

(Remarks on code availability)

Reviewer #5

(Remarks to the Author)

This paper presents two experiments in the Tacit Communication Game framework (where participants combine goal-directed and communicative movement to reach a personal goal while signalling a second location to a partner), and a computational model of this game using a surprise-based framework where signallers construct a movement path which signals goal for the receiver using surprising moves (moves with low prior probability, i.e. moving backwards or away from the eventual goal). The surprise model fits well to participant data, including pupil dilation and EEG data.

The experiment and analysis looks well conducted to me (although I have very little expertise in EEG in particular so I will defer to other reviewers on that) and the model is well explained and quite intuitive. I found the paper very interesting and exciting and would like to see it published. My only real comment concerns the connection between this work and preceding work from the cognitive science community using similar theoretical notions, experimental methods and related modelling paradigms - I think the paper could be substantially strengthened by drawing out these parallels.

The idea that speakers track the statistical properties of their productions and potentially use those statistical properties to facilitate interpretation is well established, e.g. in the form of the Uniform Information Density hypothesis (works by Roger Levy on this topic would provide a good starting point). Making this parallel explicit early would highlight how these same principles explain "regular" language use (in even more mundane contexts than the humour example you give).

Second, there is a literature on iconic communication beyond the tacit communication game that should be acknowledged and can be integrated. This includes papers presenting experiments where participants solve communicative tasks in novel communicative mediums by exploiting the iconic potential of the representational medium they are provided (see e.g. work

by Tessa Verhoef on communicating time using spatial movement, work by Bruno Galantucci and Gareth Roberts on other visual communication tasks, Marieke Schouwstra and Yasamin Motamedi on improvised gestural communication, etc etc). While it would be nice to see a broader range of such studies mentioned and if possible integrated into your framing, there are some *very* similar movement-based communication paradigms that definitely need to be mentioned, and which fit very nicely within your framework. In particular, Scott-Phillips et al, 2009, "Signalling signalhood", <https://doi.org/10.1016/j.cognition.2009.08.009>, is clearly relevant in that it involves an experiment in which participants produce flagrantly non-goal-directed movements to signal to their partner; more recent work by Julian Jara-Ettinger (e.g. most obviously <https://escholarship.org/content/qt05z6b1wh/qt05z6b1wh.pdf>, also <https://doi.org/10.1016/j.cognition.2023.105524>, but I think he has numerous relevant papers) use similar tasks and include computational models which are very similar in spirit to those you propose here; it would be fantastic to see these acknowledged, similarities and differences between the frameworks outlined.

Finally, I would like to see how your model integrates with the Rational Speech Act model (developed by Mike Frank and Noah Goodman). In particular, you only really offer an intuitive explanation of why signallers might prefer surprising movements in their messages; as I understand it, under the RSA model this would be understood in terms of the signaller designing a signal which maximises the posterior probability of a listener interpreting their movement as being directed towards the intended goal, i.e. $p(\text{goal}|\text{signal})$, the sender inverts this to get $p(\text{signal}|\text{goal})$. This sounds very similar in spirit to the Belief-Based Model which you touch on in the supporting materials, but the implementation looks somewhat different. The main thing for me is that the RSA or something potentially similar explains *why* participants would use surprising moves, from a receiver perspective - it might also allow you to model and capture the receiver success rates from your communication game, which at the moment are left un-modelled but which fall quite naturally into the picture from the RSA perspective.

Additional minor points:

Figure 1 could be better explained in the caption, I had to do quite a lot of work to infer what was going on. In particular, in figure A, please provide a legend explaining what the circle and cross symbols mean; in 1d i really could not see how that message profile was decomposed into the sequence of movements given, perhaps because I don't understand the implicit facing of the circle character icon.

Table 1, the scientific notation for PEP values is not very user-friendly and this level of precision is probably not required!

Statistical analysis of pupil response - I wasn't sure why binning of the surprise level was required, rather than treating it continuously as for the EEG data, but in any case I think you need a by-participant random slope for surprise level in your model here (equation 1 on page 21).

That's it. As I said, I really like the paper, and I think integrating it with these existing literatures will maximise the readership and the interest.

(Remarks on code availability)

Version 1:

Reviewer comments:

Reviewer #1

(Remarks to the Author)

We thank the authors for the careful revisions of the original manuscript, which we believed have greatly improved their paper. We are very satisfied with their response to our queries, and look forward to seeing this work published in Nature Communications.

(Remarks on code availability)

Reviewer #2

(Remarks to the Author)

The authors have addressed the majority of my concerns and I think the manuscript is much improved.

I remain somewhat less convinced by the EEG analysis. What the authors refer to as N300 and N400 don't really align with those components based on the amplitude patterns and topography. N400s for instance are typically centro-parietally distributed. Also the peaks appear to be more positive for high surprise events which is the reverse of the expected pattern. I think it would be somewhat more reasonable to interpret this as a prolonged P300. In any case, I think it would be sufficient to temper the conclusions about the EEG results in the main text.

As a more general point, there are a few compelling analyses and figures that appear in the response letter but not the main

text. I would recommend adding at least a footnote to briefly provide the results of these analyses since many readers may have the same questions that I had as a reviewer (e.g., participants' strategies in the game, the role of step number imbalances).

Finally, I wasn't able to look at the new code because the repo appears to have been made private now.

(Remarks on code availability)

I wasn't able to access the repository. I think maybe it has been made private since the previous submission.

Reviewer #4

(Remarks to the Author)

(Remarks on code availability)

Reviewer #5

(Remarks to the Author)

I liked the first draft of this paper, and the authors have responded appropriately to my suggestions, so I am happy to see it published. My one remaining quibble is that I think they overstate the mismatch between their approach and the conventional one from linguistics, e.g. on lines 491-501 where they say "The key insight of this study is that a framework of expectation violation can serve as a mechanism for signaling goal orientation in human communication. ... This approach contrasts with traditional communication models that rely on predictability and shared linguistic cues." I think a real strength of their experiment is that it provides a beautifully controlled environment for studying how violation of expectation is communicative, but this has long been observed for natural language too, where the prior expectations are based on knowledge of a conventional grammar and assumptions about speaker efficiency. For example, in the classic work on implicature by Laurence Horn (e.g. reviewed in <https://doi.org/10.1002/9780470756959.ch1>), it is observed that being more verbose than necessary can convey additional information, so if i say "I got the car to start", that conveys additional information on top of "I started the car" precisely *because* it's unnecessarily verbose and violates the expectation you should have that I shouldn't be aimlessly verbose. But that tweak notwithstanding, I think this is an impressive and exciting paper that I'd like to see published.

(Remarks on code availability)

made.

Dear reviewers,

Thank you very much for your careful review and thoughtful comments and suggestions on our manuscript, “Communication with Surprise – Computational Principles of Goal Signaling in Novel Human Interactions.” We appreciate your feedback and have found it very helpful in improving our work. Below, we address each of your comments and outline the changes we've made to the manuscript.

Reviewer #1:

Summary

This is an interesting and carefully conducted study, including a behavioral experiment ran on two separate groups of human participants, which was used to collect communicative signals, pupil dilation (extracted from eye-tracking) and EEG responses. In addition, the authors built a surprise model that predicted human responses with higher accuracy than degenerate versions of the same model (Movement and State models only). The results of the study confirm that, when deprived of a communicative system (be that language, gesture, or sign), humans can rely on universal laws of physics and expectancy violations (the key contribution of this work) to communicate a message to a receiver.

Key Results

The study explores how people instinctively turn to the fundamental principles of the physical world to overcome communication barriers. For this, they investigate how participants develop novel strategies such as movement kinetics for conveying messages in the Tacit Communication Game, without relying on common linguistic signals. They find that messages generated by the Surprise computational model—based on information-theoretic surprise—closely resemble human Sender messages and positively correlate with Receiver's physiological and neural responses. These findings enrich our understanding of surprise as a non-verbal communication tool, elucidating its computational and neural mechanisms.

Summary of Strength

Validity: The study is carefully designed, the experimental setup reasonable, the analysis methodologically sound, and the results are both interesting and convincing. **Significance:** 1) The Tacit Communication Game, where a Sender seeks to convey messages by defying the Receiver's expectations, has been used in non-verbal communication research for some time, yet the use of computational models in this area has been rather limited (as far as I know, the traditional belief-based models are unrealistic for TCG). Thus, from the modeling side, this

paper pushes the field in a novel direction by proposing a computational model that detects the Sender's goal from movement patterns to explain the Receiver's understanding of the signal, plus their physiological and neural responses during non-verbal communication. This movement-based innovation is quite impressive and may generate and confirm new hypotheses about forming new communicative strategies without relying on common language. 2) From the theory side, the work nicely shows that prediction errors are more than a teaching signal improving future expectations, but also encourage efficient communication. **Data, Methodology, and Analytical approach:** While I lack specific expertise in modeling the Receiver's physiological and neural responses during non-verbal communication using surprise models, my experience with agent-based modeling in communication games allows me to confirm that the data, methodology and analytical approach in this work are sound and solid. A few suggestions are made in the section below.

Response: We thank the reviewer for the positive feedback on our modeling approach. We appreciate the kind words and are glad to hear that the reviewer found our work valuable.

Major Suggestions

(1) Motivation/Contribution of this paper: The significance of this paper lies in its pioneering computational framework, which contributes a theory of surprise as a potential non-verbal communication tool. While the results of this work are interesting and solid, its current presentation falls short of fully conveying its bold contributions. Therefore, there is ample room for further refinement and elaboration to underscore the innovative nature of this research.

Response: We thank the reviewer for this positive and thoughtful feedback. We found the comments very helpful and have made every effort to address them thoroughly. Now, in response to the suggestion to enhance the presentation of our paper's contributions, we have made significant revisions to the introduction to better highlight the innovative nature of the computational model we propose. Briefly, in the introduction, we now describe predictability as a key mechanism of verbal communication, contrast it with situations requiring unconventional strategies, and demonstrate how our computational framework unifies these contexts through expectation violation. For a detailed view of the changes, please refer to the quoted text from the revised manuscript below.

Lines 27-58 from the revised introduction

“Effective communication relies on the ability of the Sender to design messages that clearly convey intentions, and the ability of the Receiver to accurately interpret these messages. In verbal communication, this is achieved as speakers and listeners track the statistical properties of language to optimize communication, often selecting the most predictable outcomes to facilitate understanding¹. The Rational Speech Act model, for instance, posits that speakers optimize utterances to minimize listener's surprise while

maximizing the likelihood of accurate interpretation by the listener². Similarly, the Information Density Hypothesis suggests that speakers manipulate syntactic structures to maintain a consistent flow of information, by aligning their linguistic choices with the statistical regularities of speech³.

However, this predictability, seen in natural language processing, is not always maintained in communication. In many instances, communication takes on unconventional forms to achieve a specific goal. For example, in signaling behavior where the Sender needs to strategically direct the attention of the Receiver, it has been demonstrated that inefficiencies can signal particular intentions⁴, objects, even those not intrinsically communicative like chairs or ropes, can convey social meanings through their contextual arrangement⁵, and non-verbal cues can embody complex concepts like time⁶. These examples, while diverse in communicative strategies, highlight a common theme: the strategic deviation from expected communicative patterns to achieve greater clarity and intentionality in communication. These deviations are particularly effective because they contrast sharply with anticipated behaviors, thereby capturing attention and signaling intentions more distinctly.

Here, we propose a unified framework of expectation violation as a mechanism for the signaling function of human communication. This approach entails two main mechanisms. First, when faced with situations lacking a common language, humans instinctively revert to more basic, universal properties of the physical world (e.g., movement direction or velocity) to form a shared understanding. It is not about discovering new universal principles; rather, it is about repurposing universally understood properties to build a new common communicative system through repeated social interactions. Second, once these common grounds are established, they serve as the prior expectations from which Senders can intentionally deviate. These deviations are used to create surprise, which directs the Receiver's attention to essential information and transforms into communicative acts.”

(2) From a theoretical perspective, this work unfortunately falls short of shedding light on real, verbal communication. The authors do not even attempt to draw any conclusions from their findings that could apply to verbal communication. I would therefore encourage the authors to engage in such discussion in order to highlight the value of their study.

I recommend the authors to contrast their notion of information-theoretic surprise (which was the driver of their results) to that of predictability in language processing (which is known to be a driver of verbal communication).

Ryskin, R., & Nieuwland, M. S. (2023). Prediction during language comprehension: what is next? *Trends in Cognitive Sciences*.

For a direct comparison of these two forces in human communication (surprise vs predictability) see also:

Rohde, H., & Rubio-Fernandez, P. (2022). Color interpretation is guided by informatively expectations, not by world knowledge about colors. *Journal of Memory and Language*.

Another point of comparison with the referential communication literature (including modeling work) is the kind of inferences that Receivers can derive from Sender's choices (e.g., an inefficient choice tends to be interpreted as an ostensive signal, the same way that participants did in this study).

Royka, A., Chen, A., Aboudy, R., Huanca, T., & Jara-Ettinger, J. (2022). People infer communicative action through an expectation for efficient communication. *Nature Communications*.

Jara-Ettinger, J., & Rubio-Fernandez, P. (2021). Quantitative mental-state attributions from linguistic events. *Science Advances*.

One last interesting point of comparison between the Surprise Model reported in this manuscript and previous computational models of referential communication is the notion of Reward as a bias for shorter expressions. For a recent discussion of different cost functions in referential communication, see:

Jara-Ettinger, J., & Rubio-Fernandez, P. (2022). The social basis of referential communication: Speakers construct physical reference based on listeners' expected visual search. *Psychological Review*.

The authors are not requested to cite these works. I am just suggesting relevant literature, but they are free to review other papers and get into the same issues.

Basically, I think the Discussion section should address the following questions: what do the results of this experiment and the computational models tell us about human communication, by and large? If the principles underlying these results and models are universal, how do they transfer to linguistic communication?

Response: This is a very good point. We acknowledge that our initial discussion did not sufficiently draw parallels with established literature and thank the Reviewer for pointing it out.

First, we incorporated key theories of verbal communication in the introduction, as quoted in lines 27-35 in the previous response. We then contrasted the role of surprise with the

predictability inherent in these theories in the discussion. Here is the paragraph from the Discussion:

Lines 500-513 from the revised discussion

“This approach contrasts with traditional communication models that rely on predictability and shared linguistic cues^{2,3}. These are more naturalistic scenarios where reliance on universal knowledge is unnecessary because the presence of shared language enables predictability to serve as the primary driver of language comprehension. For example, listeners and readers often use contextual information to predict upcoming linguistic input¹, and predicting this input facilitates faster understanding of the message⁷. This predictability of a word in the context of a sentence is described by quantities from information theory, namely the KL divergence (or Shannon surprise)⁸. This theoretical framework partially aligns with our Surprise model, which also uses Shannon surprise to create messages that are informative for the Receiver in finding their goal, provided they can decode the surprise by evaluating steps with respect to the priors, constructed from universal principles of movement kinetics. However, unlike traditional models that focus on minimizing surprise to enhance comprehension, our framework maximizes surprise. This approach is particularly crucial in the absence of a shared language base, where maximizing surprise helps convey information more effectively.”

Next, we draw parallels to the referential communication literature, where a similar mechanism of communication is observed.

Lines 515-524 from the revised discussion

“Similar principles have been observed in other referential communicative situations as well. Participants interpret communicative actions through an expectation for efficient communication, often perceiving inefficient actions as intentional when they are unexpected⁴. This supports our observation that unexpected actions in the TCG are often the most communicatively effective. Additionally, research indicates that when interpreting color adjectives, people prioritize context-specific informativity over general world knowledge⁹. In experiments, participants were more likely to choose a less typical yellow object, such as a chair, over a more common one, like a banana, when the color adjective played a crucial role in identification. This finding suggests that, in language processing, context-specific expectations can override general assumptions based on world knowledge.”

At last, we answered the reviewer’ concluding questions in the discussion (lines 540-548): What do the results of this experiment, and the computational models tell us about human communication, by and large? If the principles underlying these results and models are universal, how do they transfer to linguistic communication?

Lines 526-534 from the revised discussion

“Comparing these two mechanisms of communication—reducing surprise through predictability and increasing surprise by deviating from expectation—reveals that human communication fundamentally involves managing expectations. While using surprise is more typical in novel communication without a shared language, it can also be employed in verbal communication. Introducing unexpected elements in verbal interactions can enhance engagement, memorability, and effectiveness. Examples include jokes, which rely on unexpected punchlines to elicit laughter and engage the audience¹⁰, and prosody, which uses variations in pitch, tone, and rhythm to maintain interest and enhance comprehension¹¹. These two strategies of managing expectations demonstrate the adaptability of human communication across various contexts.”

Minor Suggestions

(1) Discussion of AI models: The Discussion section attempts to contextualize the impact of this work within the recent advancements of large language models, proposing the consideration of a “surprise selection algorithm” that occasionally opts for less expected alternatives. While I appreciate the authors’ intention, caution is warranted regarding two aspects: 1) Clarification is needed on how the action selection is generalized to word selection (e.g., referencing Shain et al. 's work). 2) Large language models do not always opt for the word with the highest probability; instead, various hyperparameters and techniques are employed to prevent the model from consistently selecting the most probable word, including top-k sampling or temperature scaling.

Response: In response, we have clarified the points the reviewer raised:

1. **Generalization from Action to Word Selection:** In response to reviewer’s inquiry about generalizing action selection to word selection, we have compared the mechanism of action selection to word selection in the discussion, quoted in the previous answer (503-513). Briefly, the Surprise model uses information theory, specifically Shannon surprise, to create messages that help the Receiver achieve their goal by decoding unexpected actions. This framework also applies to natural language, where the surprisal of a word in a sentence can be described using information theory concepts like KL divergence. The difference between these two mechanisms is that one minimizes surprise to aid comprehension, while the other maximizes surprise.
2. **Large Language Models and Probability Selection:** We acknowledge the reviewer's correct observation that LLMs employ various strategies beyond simply selecting the highest probability word. Techniques such as top-k sampling and temperature scaling indeed allow LLMs to explore a wider range of word choices, thereby preventing overfitting and enhancing the diversity of generated content. In our revised discussion, we highlight that these algorithms ensure moderate variability in word selection, whereas applying the principles of our Surprise model would lead to the selection of greatly unexpected words,

thus ensuring an even greater variability in the selection process. Here is the revised text from the discussion.

Lines 594-603 from the revised discussion

“Integrating this concept with AI technologies could potentially make AI more human-like and creative. Generative AIs trained on extensive text typically analyze sets of possible words based on contextual information to ensure clarity and coherence by selecting the most probable words. Even though these selection algorithms occasionally introduce some variability through top-k sampling and temperature scaling, they only ensure moderate variability in word selection. Applying the principles of our Surprise model would lead to the selection of greatly unexpected words, thus ensuring even greater variability in the selection process. This adaptation would be a novel approach, potentially diverging from predictable paths to enhance creativity and engagement. Such an application could enhance AI’s capabilities in comedy, storytelling, or generative art, creating more engaging and unique scenarios.”

(2) Writing Clarity: The writing of this work requires significant improvement in several areas.

1) The introduction lacks motivation for developing movement kinematics models and fails to highlight their differences from traditional models, such as belief-based models. 2) In the results section, there is a need for more motivation before delving into the elaboration of results. It would also be beneficial to include a brief summary for each Results subsection. Additionally, many subheadings lack informativeness, such as "Predictive Accuracy of the Surprise Model across Different Sample" and "Model Fitting and Model Comparison." These headings should be grounded with specific results. 3) The work comprises multiple subparts, including experiments, modeling, and brain data analysis. It might be beneficial to include a single visualization in the introduction that encapsulates all these subparts.

Response: We appreciate these very helpful suggestions and agree that they indeed improve the readability and clarity of the text. Below, we briefly outline how we addressed these points in the revised manuscript.

1. **Motivation for developing movement kinetics models:** We have revised the introduction (lines 68-76) to better clarify this aspect.

“Given that TCG focuses on spatial movement and trajectories, in the Surprise model, we developed two types of priors: one based on movement kinetics, such as the expectation that a moving object will continue its path unless acted upon by an external force¹², and the other based on the Sender’s goal orientation. These priors create the core of a new common language, where trajectories along a straight line signify a direction and embody a shared expectation. Deviations from these paths thus transform into communicative acts. The Surprise model utilizes these priors to construct messages that maximize information-theoretic surprise, which quantifies the

psychological state of surprise at the Receiver's goal state. By leveraging these deviations from expected paths, communicative acts become signals with meaning through social interactions.”

2. Regarding the **comparison with Belief-Based Models (BBM)**, we have incorporated a detailed paragraph in the discussion section (lines 547-559) to address this.

“An alternative to expectation management in communication is employing the Theory of Mind (ToM). ToM models form beliefs about a partner's behavior and choose actions accordingly¹³⁻¹⁵. In the TCG context, a ToM model requires defining a belief distribution over possible states and exhaustively searching for each goal configuration, demanding extensive memory and contradicting the brain's preference for efficiency. The Belief-based Model (BBM)¹⁶ is an example of such a model that includes ToM-like mentalization, but it operates on the entire message, thus precluding step-by-step analyses. We qualitatively compare the BBM to the Surprise model by simulating message selection for all experimental goal configurations, but because of the impossibility of calculating a subject-specific likelihood the BBM was not included in formal model comparison. These simulated messages from the BBM and a Bayes factor analysis showed that the model fails to capture human message generation, suggesting ToM is not the primary mechanism in TCG communication. Detailed information about BBM structure and simulation outcomes are available in the supplementary material.”

3. To **enhance the motivation of our results section**, we have incorporated an introductory paragraph (lines 106-117) before the result sections. In this paragraph, we reiterate our study's objectives to provide a clear context for the forthcoming results.

“We explored how humans form novel communication systems without a shared language, and how deviations from expected patterns can be used to convey information in the Tacit Communication Game (TCG). This game allowed us to estimate the effectiveness of the Surprise Model in replicating human behavior and the physiological and neural correlates of surprise during non-verbal communication. Our goal was to demonstrate that humans can use basic principles of movement kinetics to create new communication norms, and that these norms can be systematically studied and modeled.

We first provide a brief overview of the TCG, followed by the model-free analysis of behavioral data. Subsequently, we introduce our computational models. Building upon this, we proceed to validate the computational models and explore physiological observations related to non-verbal communication within the TCG framework.”

4. Additionally, we added a **summary after every subsection of results**. For example, see lines 213-218 after model-free analysis of behavioral data.

“In summary, this analysis highlights how Senders adapt their communication strategies within the TCG, opting for different message types depending on the trial type. The analysis highlights the cognitive flexibility required to navigate the game’s constraints, revealing that different strategies significantly impact communication success. These insights into message usage and effectiveness are crucial for understanding the underlying cognitive processes and refining cognitive models of non-verbal communication that we will present in the next section.”

5. We also **updated the titles of the subsections within the results to provide clearer insights into the findings**. Here are some examples of these modifications: In place of “Computation Models,” we adopted “Information-theoretic surprise as communication signal,” offering a more specific descriptor. We replaced “Model fitting and model comparison” with “Surprise model accurately explains Sender’s message design,” directly reflecting the results' focus. Please note that these titles are just examples, and we have changed almost all subheadings similarly.
6. As recommended by the reviewer, we have included this figure within the introduction

Figure 1 | Overview of experimental procedure and analysis This figure depicts an experimental framework utilizing the Surprise Model within the Tacit Communication Game (TCG). The experiment begins by recording behavioral data from both Senders and Receivers as they engage in TCG. This recorded data is then used to fit the Surprise Model to estimate the individual parameters. With these parameters, the model generates communicative messages, each accompanied by message steps and

their corresponding surprise values. These values are subsequently analyzed to assess their influence on the Receiver's physiological responses, specifically through EEG and PDR measurements.

section. This figure outlines our experimental approach, starting from data collection to the prediction of physiological data.

Reviewer #2:

Remarks to the Author

This paper investigates how humans communicate information when language and gesture are not available and the signal consists of movements in a 4×4 spatial grid. Specifically, the senders need to communicate to the receivers what spatial location they need to move to while simultaneously moving to their own (different) goal location. The authors propose that senders will draw the receiver's attention to their goal by maximizing the receiver's information-theoretic surprise which is derived from their prior beliefs about how the sender is likely to move (preferring to continue on the same path than deviate from it and to move closer toward her goal with each step). They develop a computational model which simulates the sender's messages/paths and fit its free parameters to behavioral data, demonstrating impressive posterior prediction relative to simpler models. They then show that the surprisal of a given step in the message is correlated with the receiver's pupillary response as they view the message step. Finally, they show that EEG responses collected from receivers are also sensitive to the surprise in two time windows with distinct spatial distributions.

Overall, this is a very exciting paper that should be of interest to a broad cognitive science/neuroscience audience. The writing is clear, and the methods are appropriate for the questions at hand. Nonetheless, some aspects of the paper could be strengthened.

Response: We appreciate the reviewer's positive feedback. We are pleased to hear that the reviewer found the paper exciting and relevant to a broad audience.

Suggestions

(1) The title is perhaps a bit misleading. Non-verbal communication could refer to a large number of things. Often people use it to mean gesture, eye gaze, and other cues that humans use in everyday life to supplement verbal communication or when verbal communication is temporarily unavailable (e.g., when people don't want to be heard). This is quite different from

the situation studied here, in which communication is not simply constrained to be non-verbal, it is a novel communication system using motion in a 2D space.

Response: This is indeed a very good and important point. We have revised the title to better reflect the content of the research:

New Title: “**Communication with Surprise: Computational Principles of Goal Signaling in Novel Human Interactions**”

This updated title clarifies two key aspects:

1. **Goal Signaling:** It specifies that our focus is on the signaling of goals, rather than non-verbal communication more broadly.
2. **Novel Human Interactions:** It emphasizes the context of novel human interactions where there is no shared language, aligning more closely with the specifics of our study.

We believe that this title more accurately represents the focus of our paper.

(2) It would be useful to add some details about the experiments. For instance, I was curious to know the exact way in which participants were instructed to play the game and whether they had any practice with feedback. Did the instructions make any mention of the receiver's “expectations” or “deviations” in the path? Did participants instantly pick up on one of the three strategies, or did they initially fail? It would also be interesting to know how performance changed across the trials. And, though this may be beyond the scope of the current paper, did the receiver's PDR or EEG response predict more successful communication on subsequent rounds?

Response: We appreciate these very valid points raised and will address them in two parts: first, the instructions and experimental design, and second, about the participants' behavior in the game:

1. **Details on the experimental design and instructions.** We have modified the section on Experimental procedure in the Methods (lines 636-651) to answer these questions:

“Prior to the experiment, the participants were placed in the same room in front of separate screens to play the game after they had been assigned roles and given detailed standardized instructions. Instructions informed them that success in the game depended on both players reaching their respective goal locations. Specifically, the Sender was aware of both players' destinations and needed to guide the Receiver to his undisclosed goal location. Additionally, both players were told each movement cost a score, and the bonus at the end of the game would be calculated based on the sum of the score from all the trials. We clarified that movement on the grid was restricted to horizontal or vertical directions. Importantly, there were no practice rounds offered, allowing us to observe the whole spectrum of behavior from the moment participants were exposed to the game rules until the end. The instructions deliberately avoided

Editorial Note: Figure 2 was created using Matlab R2023a.

mentioning any specific strategies like managing “expectations” or “deviations,” to prevent biasing participants towards any predetermined strategies. Our goal was to capture a wide range of natural communication behaviors as players adapted to the game’s requirements. The TCG game consisted of 120 trials. In every trial, players had to solve different goal configuration problems. A goal configuration consists of three elements: 1) the starting location in one of 4 middle states of the 4×4 game board (same for both players), 2) the Sender’s goal location, and 3) the Receiver’s goal location.”

2. **Participant’s behavior in the game.** To address the reviewer’s questions about participant strategies and performance changes throughout the trials, we refer to the figures provided below. Participants exhibited an initial average accuracy of 0.7 across all participants and showed a steady improvement, reaching higher levels of precision towards the latter trials (Fig. 2a). This pattern suggests a gradual optimization of strategies as participants adapted to the game mechanics and the receiver's responses. As shown in Figure 2b, different message types—Enter-Exit, Wiggly, and Pass-By—varied in effectiveness. Enter-Exit consistently yielded the highest accuracy, followed by Wiggly, while Pass-By showed lower effectiveness, particularly in the initial trials. This indicates that some strategies were more intuitive or effective for conveying the intended message, with Enter-Exit being the most successful from the onset.

Figure 2 | Participant’s behavior in TCG

(a) Mean accuracy over the trial blocks: Initial average accuracy of 0.7 improved steadily, indicating strategy optimization as participants adapted to the game. **(b) Mean accuracy by message**

type: Enter-Exit yielded the highest accuracy, followed by Wiggly. Pass-By was less effective, especially in early trials, suggesting some strategies were more intuitive. **(c) Frequencies of message types for 4 exemplary participants over the trial blocks:** Participants 3 and 9 initially used various strategies, but later focused on Enter-Exit or Wiggly. Participant 5 consistently mixed strategies, while Participant 10 exclusively used Enter-Exit. Strategy choices were influenced by receiver performance and trial type.

Figure 2c exemplifies the diversity in strategic choices by featuring several individual participants. For example, subjects 3 and 9 experimented with multiple strategies initially, but predominantly adopted either Enter-Exit or wiggly strategies as the game progressed. In contrast, subject 5 utilized a mix of strategies throughout the game. Subject 10 employed the Enter-exit strategy universally. Mainly, the strategic choice was influenced by both the receiver's performance and the trial type—direct or indirect—prompting participants to adjust their approach trial-by-trial.

Lastly, we would like to address the question regarding the **predictive power of physiological measures** on communication success in subsequent rounds.

Given the current experimental design, we do not expect to obtain predictive results from this data, and here we will explain why. The experiment involved various message types, such as Enter-Exit, Pass-By, and Wiggly messages. Enter-Exit messages, which may result in higher PDR values indicating heightened surprise, would not necessarily predict success in subsequent trials. This is because subsequent trials could involve Pass-By messages, which are inherently more difficult. Therefore, high PDR during a trial using Enter-Exit messages does not necessarily correlate with success in the following trial using different strategies. The variability in message types throughout the experiment highlights the challenge of using PDR responses to predict communication success in subsequent rounds. Given this variability, we do not expect consistent predictive results from the current experiment. Future experiments that control the sequence of message types could potentially address this question more effectively.

Additionally, if 'round' refers to the entire trial, averaging the Pupillary Dilation Response (PDR) across all steps within that trial could introduce significant variation and potentially obscure meaningful patterns. This is due to the variance in surprise levels across different steps of the message, as well as the varying number of steps per message.

(3) The Belief-Based Model comes out of nowhere on page 12 and doesn't even get a reference. Perhaps something got cut from an earlier section of the manuscript, but it would be helpful to explain what it is and why it is worthwhile to compare it to the Surprise model. It is actually a strength of this work that there is an alternative theoretical account which does not capture the behavioral data as well, and this should be made clear in the main body of the paper.

Response: We have moved the discussion about BBM from the results to the discussion (lines 547-559) where we compare our Surprise Model with general theory of mind models and specifically discuss the limitations of the BBM¹⁶ in the context of TCG and why we did not include the results in the main text.

“An alternative to expectation management in communication is employing the Theory of Mind (ToM). ToM models form beliefs about a partner’s behavior and choose actions accordingly¹³⁻¹⁵. In the TCG context, a ToM model requires defining a belief distribution over possible states and exhaustively searching for each goal configuration, demanding extensive memory and contradicting the brain's preference for efficiency. The Belief-based Model (BBM)¹⁶ is an example of such a model that includes ToM-like mentalization, but it operates on the entire message, thus precluding step-by-step analyses. We qualitatively compare the BBM to the Surprise model by simulating message selection for all experimental goal configurations, but because of the impossibility of calculating a subject-specific likelihood the BBM was not included in formal model comparison. These simulated messages from the BBM and a Bayes factor analysis showed that the model fails to capture human message generation, suggesting ToM is not the primary mechanism in TCG communication. Detailed information about BBM structure and simulation outcomes are available in the supplementary material.”

(4) A few aspects of the statistical analysis of PDR seemed like they required some explanation. Why was surprise, a continuous variable, binned into a 3-level categorical variable? The continuous version seems more appropriate. Also, it would be appropriate to include random slopes for the (continuous) effect of surprise for each participant in the mixed-effects models. Was this omitted due to convergence issues? (There are workarounds for that, e.g., using allFit()).

Response: We have incorporated these very good suggestions in a revised analysis using surprise as a continuous variable. We included random slopes for participants in addition to random intercepts to account for individual variability in responses. These new results are quoted below.

Updated text in the revised results section (lines 381-411):

“Previous studies have demonstrated a close link between pupillary dilation responses (PDR) and violations of expectations (i.e., a surprise)¹⁷. We therefore hypothesized that the step-by-step surprise values calculated by the Surprise model would correlate significantly with the Receiver's PDR while observing the Sender's messages. To test this, we devised a model-informed analysis, in which we used the step-by-step surprise values from the Surprise model as a predictor for the participant's step-by-step PDR and used a mixed-effects model as implemented in the lmer package in R to test this correlation¹⁸. In the model, we incorporated a fixed effect for the task condition—a continuous variable representing surprise values. We also included random intercepts

and slopes for each participant. The mean PDR change of message steps served as the predicted variable.

The results revealed that the baseline-corrected PDR (w.r.t. to a pre-stimulus baseline of -200ms to onset) exhibited a significant effect for surprise ($F(1,20)=60.20$, $p<.001$, $\eta^2=0.75$). Specifically, the analysis demonstrated a significant relationship between continuous surprise values and PDR, as visualized in Figure 6a. The fixed effect estimate for surprise (0.02) indicates that for each unit increase in surprise, the baseline-corrected PDR increases by approximately 0.02 units. This positive linear relationship confirms that greater surprises induce more pronounced physiological responses, as measured by pupillary dilation.

Figure 6 | Model-based analysis of PDR data

(a) Predicted PDR: The graph shows the predicted baseline-corrected PDR as a function of model-derived surprise values. The relationship demonstrates a significant linear increase in PDR with increasing surprise values. The shaded area represents the 95% confidence interval **(b) Surprise levels during observation:** As the message steps are observed by the Receiver, varying levels of surprise associated with different steps are expected. Certain steps are more surprising than others.

In summary, we investigated how surprise, predicted by the Surprise model, correlates with pupillary dilation responses (PDR). The analysis shows a significant correlation between surprise and PDR. This confirms that surprise effectively triggers measurable physiological responses, underscoring its impact in non-verbal communication settings.”

Updated text in the revised methods section (lines 707-717):

“We investigated the effect of surprise values generated by the Surprise model on the Receiver’s PDR, using a mixed-model analysis implemented in the lme4 package in R

version 3.6.1, with the parameters estimated through maximum likelihood estimation¹⁸.

In the model, we included a fixed effect for surprise values, and a random intercept and slope for each participant:

$$PDR \sim surprise + (1 + surprise | sub) \quad (1)$$

To test the significance of the fixed effects, we used the likelihood ratio test (LRT) implemented in the lme4 package in R¹⁸.”

(5) The EEG analysis, though a novel aspect of the study, was less compellingly presented. At a high level, the inferences that the fronto-central cluster reflected ACC activity related to prediction error and the frontal cluster reflected prefrontal activity related to cognitive control are highly speculative. They would be perhaps bolstered by attempting to tie these spatio-temporal clusters to known EEG components with well-described functional properties (e.g., P300). Otherwise, the conclusions about signal sources should be qualified. Also, it wasn't clear to me what to make of the analysis where the data were split by message type. Couldn't the strong relationship in Enter-Exit be mostly explained by the fact that there were more trials of that type?

Response: This is a very valid point, and we thank the reviewer for pointing it out. To strengthen our inferences regarding the neural sources of surprise processing, we conducted additional analyses that directly address the reviewer's concerns. These included:

1. ERP Analysis for Low and High Surprise Events: We computed event-related potentials for the top 20% (high surprise) and bottom 20% (low surprise) of surprise trials to tie the observed spatio-temporal clusters to known EEG components. As depicted in Figure 7a, the ERPs for high surprise events showed distinct peaks at approximately 300 ms (N300) and 400 ms (N400) post-stimulus, which are components typically associated with cognitive processing and error detection. These peaks were not present for the low surprise events.
2. Source Localization Analysis: We performed source localization analysis using individual participants' structural MRI images and individually recorded electrode positions to identify the neural substrates of the EEG signals associated with surprise. The source localization results, as shown in the attached figure (Figure 6b), revealed a source activation in regions close to the supplementary motor area (SMA) and the dorsal anterior cingulate cortex (dACC), two regions known to be involved in motor planning as well as in error detection and conflict monitoring respectively.

Editorial Note: Figure 3 was generated using the FieldTrip toolbox, which is licensed under CC BY-SA 4.0 (<https://creativecommons.org/licenses/by-sa/4.0/>).

Figure 3 | Additional analysis of surprise processing (a) Receiver's ERPs for low and high surprise events: The ERPs show distinct N300 and N400 components for high surprise events, indicating cognitive processing and error detection. **(b) Source localized activation:** Activation in the supplementary motor area (SMA) and dorsal anterior cingulate cortex (dACC) is observed, providing a neural basis for the EEG clusters identified.

We have updated Figure 6 in the revised manuscript to include the source localization results. Additionally, we have reported the results of the source localization analysis in the Results section (lines 435-439), described the procedures in the Methods section (lines 751-780), and discussed the implications in the Discussion section (lines 571-586). Below is the corresponding paragraph from the discussion:

“Sensor-level analysis of the neural encoding of surprise in the EEG data revealed two prominent time-space clusters over frontal and fronto-central electrodes. Source localization analysis identified a dipole, the potential neuronal origin of these clusters, in the region encompassing both the Supplementary Motor Area (SMA) and the dorsal anterior cingulate cortex (dACC). The first cluster, appearing around 400ms after the onset of the communication step, could be linked to the Receiver mentally simulating the Sender's intended motor actions while observing the message. This likely involves the SMA, which is typically engaged in motor planning¹⁹. After simulating the Sender's motor plan, the Receiver can detect deviations from the underlying priors. We associate this detection with the second cluster, potentially originating in the (dorsal) anterior cingulate cortex (ACC), known for its role in error detection. The ACC integrates diverse information and guides decisions to align with goals and reward representations^{20,21}, playing a crucial role in linking actions to outcomes and evaluating their worthiness^{22,23}. Recent studies also show that the ACC discerns others' intentions and adjusts perceptions based on new evidence^{24,25}. Our findings align with

this perspective, suggesting that the second cluster, presumably reflecting ACC activity, is associated with the Receiver’s ability to detect surprise as intentionally designed by the Sender²⁶.”

Regarding the **Analysis by Message Type**: We understand the concern about the potential influence of varying trial counts on the observed relationship between surprise levels and EEG signals.

(a) Analysis with all steps

(b) Analysis with balanced number of steps

Figure 4 | Comparison of regression coefficients with all steps vs. balanced steps across message types. (a) displays the analysis using all steps, with the bar graph on the left showing the total number of steps for each message type and the regression coefficients over time for each message type on the right. (b) presents the analysis with a balanced number of steps, confirming that the number of selected steps is equal across message types and showing regression coefficients over time that mirror the pattern observed in Panel (a).

To address this, we balanced the number of steps (trials) generated from each message type to ensure that the observed effects were not driven by disparities in trial counts. This involved selecting an equal number of steps across all message types. Specifically, we conducted the analysis by subsampling the steps to match the message type with the least number of trials, thereby maintaining a balanced dataset. The number of analysis trials decreased significantly as

we limited the dataset to match the smallest number of trials, which was approximately 1,400 for the Pass-By message type.

We then compared the results from the original dataset, which included all steps, to those from the balanced dataset. As shown in Figure 1, the key findings remained consistent, indicating that the observed effects are not solely influenced by the number of trials.

(6) In the description of the priors, the movement priors are not very intuitive. It might help to visualize how they pattern. I'm imagining maybe a raster plot with $p(f)$ on the x-axis, $p(l)=p(r)$ on the y-axis, and $p(b)$ represented as hue, though there might be a better way to do this.

Response: We appreciate the reviewer's feedback. In the original manuscript figure, we visualized the movement priors for a single step (Figure 3a in the original manuscript and here in the response letter figure 5a). The new subfigure (Figure 5b) illustrates how these priors shift as the agent moves around, with orientations defined relative to the previous location. We hope this provides a clearer understanding of the dynamic changes in movement probabilities.

(a) Movement Priors For One Step (b) Shift in Movement Priors with Message Orientation

Figure 4: Movement Prior Shifts with Message Orientation

(a) Movement Priors for One Step: Movement probabilities ($P(F)$, $P(L)$, $P(R)$, $P(B)$) shown with a color gradient, where the orange circle marks the initial position. **(b) Shift in Movement Priors with Message Orientation:** Illustrates how movement priors shift as the agent moves, with orientations defined relative to the previous location over three steps.

(7) I applaud the authors for making much of the data and code available on GitHub. However, from a brief look, it didn't seem like the actual implementation of the model(s) was available in the repo, only the result of model fitting. If that's not the case, perhaps it could be more clearly documented.

Response: We apologize for oversight regarding the inclusion of the actual implementation of the model, besides the model fitting code. To address this, we have now added a new script titled “Surprise Model” to the main repository. This script includes the full implementation of the model, with step-by-step explanations of its operations. We hope these additions will make the repository more useful.

(8) Finally, the discussion missed the opportunity to comment on how the principles of the Surprise model would generalize to other non-grid-based domains and how we might think about scaling this up to explain more naturalistic human communication behaviors. Moreover, it would be interesting to draw comparisons to other frameworks for communicative decisions, such as the Rational Speech Acts framework (Goodman & Frank, 2016), which have focused on situations where speakers try to minimize listeners' surprise.

Response: This is a very good point, and we acknowledge that our initial discussion did not elaborate on these aspects. We have addressed this in the revised manuscript.

First, in the Discussion (lines 500-513), we contrasted communicating with surprise with communicating through predictability by discussing the main theories of naturalistic human communication, including the RSA². Here is the quoted text:

“This approach contrasts with traditional communication models that rely on predictability and shared linguistic cues^{2,3}. These are more naturalistic scenarios where reliance on universal knowledge is unnecessary because the presence of shared language enables predictability to serve as the primary driver of language comprehension. For example, listeners and readers often use contextual information to predict upcoming linguistic input¹, and predicting this input facilitates faster understanding of the message⁷. This predictability of a word in the context of a sentence is described by quantities from information theory, namely the KL divergence (or Shannon surprise)⁸. This theoretical framework partially aligns with our Surprise model, which also uses Shannon surprise to create messages that are informative for the Receiver in finding their goal, provided they can decode the surprise by evaluating steps with respect to the priors, constructed from universal principles of movement kinetics. However, unlike traditional models that focus on minimizing surprise to enhance comprehension, our framework maximizes surprise. This approach is particularly crucial in the absence of a shared language base, where maximizing surprise helps convey information more effectively.”

Second, we demonstrated that the mechanism of expectation violation drives behavior in other non-grid domains as well. Here is the quoted text from the revised discussion (lines 515-524).

“Similar principles have been observed in other referential communicative situations as well. Participants interpret communicative actions through an expectation for efficient communication, often perceiving inefficient actions as intentional when they are unexpected⁴. This supports our observation that unexpected actions in the TCG are often the most communicatively effective. Additionally, research indicates that when interpreting color adjectives, people prioritize context-specific informativity over general world knowledge⁹. In experiments, participants were more likely to choose a less typical yellow object, such as a chair, over a more common one, like a banana, when the color adjective played a crucial role in identification. This finding suggests that, in language processing, context-specific expectations can override general assumptions based on world knowledge.”

And lastly, we showed that a similar mechanism of communication with surprise can be used in more naturalistic forms of communication, with the examples of humor and prosody (lines 526-534 in the discussion).

“Comparing these two mechanisms of communication—reducing surprise through predictability and increasing surprise by deviating from expectation—reveals that human communication fundamentally involves managing expectations. While using surprise is more typical in novel communication without a shared language, it can also be employed in verbal communication. Introducing unexpected elements in verbal interactions can enhance engagement, memorability, and effectiveness. Examples include jokes, which rely on unexpected punchlines to elicit laughter and engage the audience¹⁰, and prosody, which uses variations in pitch, tone, and rhythm to maintain interest and enhance comprehension¹¹. These two strategies of managing expectations demonstrate the adaptability of human communication across various contexts.”

Reviewer #3:

This work addresses a very relevant topic for the study of communication and, more generally, the regulation of relationships mediated by surprise and unexpected outcomes. Through the use of neurophysiological measures (in particular EEG and pupillary dilation responses) and the application of TCG, the authors detected the presence of specific cortical networks responding to the experimental condition. The work is well articulated, handsomely solid and accurate in its discussion phase on the results. Some observations may serve to improve the current form of the paper:

Response: We appreciate the reviewer’s positive remarks and have made every effort to thoroughly address all the comments and suggestions below.

Suggestions

(1) The link between the theoretical introduction which largely refers to the communicative phenomena which testify to the effect of the violation of expectations (for example humor) and the present experimental setting, based on an abstract game, should be better specified, also for the purpose to understand its possible applications and implications for real communication dynamics.

Response: We thank the reviewer for this suggestion. We understand the necessity of clarifying the link between the theoretical introduction and the present experimental setting of the Tacit Communication Game (TCG). Below, we explain these connections in more detail.

In the revised version of the introduction, we first propose our framework in one paragraph after contradicting its main principles with the principles of verbal communication. Here is the revised paragraph (lines 49-58):

“Here, we propose a unified framework of expectation violation as a mechanism for the signaling function of human communication. This approach entails two main mechanisms. First, when faced with situations lacking a common language, humans instinctively revert to more basic, universal properties of the physical world (e.g., movement direction or velocity) to form a shared understanding. It is not about discovering new universal principles; rather, it is about repurposing universally understood properties to build a new common communicative system through repeated social interactions. Second, once these common grounds are established, they serve as the prior expectations from which Senders can intentionally deviate. These deviations are used to create surprise, which directs the Receiver's attention to essential information and transforms into communicative acts.”

Next, we discuss (lines 60-76) how the TCG offers a controlled setting to test these theoretical principles/proposed mechanisms and how the Surprise model embodies these principles. Here is the revised text from the introduction:

“To substantiate this proposed framework, we developed a computational model known as the Surprise model and tested its effectiveness within the context of the Tacit Communication Game (TCG). TCG is an experimental task designed to elicit novel communicative messages through spatial trajectories on a square game board²⁷⁻³⁰. Each player, known as the Sender and the Receiver, has distinct goals, but only the Sender is aware of both positions. The Sender's objective is to create a trajectory (the “message”) that begins at her starting point and leads toward her goal location, while simultaneously indicating the Receiver's goal location (see Fig.1).

Given that TCG focuses on spatial movement and trajectories, in the Surprise model, we developed two types of priors: one based on movement kinetics, such as the expectation that a moving object will continue its path unless acted upon by an external

force¹², and the other based on the Sender's goal orientation. These priors create the core of a new common language, where trajectories along a straight line signify a direction and embody a shared expectation. Deviations from these paths thus transform into communicative acts. The Surprise model utilizes these priors to construct messages that maximize information-theoretic surprise, which quantifies the psychological state of surprise at the Receiver's goal state. By leveraging these deviations from expected paths, communicative acts become signals with meaning through social interactions."

(2) the reason why this task (TCG) was specifically chosen compared to other more "realistic" tasks should be better specified.

Response: The choice of the Tacit Communication Game (TCG) as our experimental task is motivated by several key factors that align with the objectives of our study. As outlined in the Introduction and further detailed in the Results and Methods section, (1) TCG gives us the opportunity to observe how the novel communicative messages are created by the sender and how these messages are interpreted by the receiver. This is critical because our goal is to observe how people communicate when there is no common-shared language, and how people in this situation revert to more universal knowledge such as principles of movement kinetics. (2) As detailed in the revised lines (68-76), quoted above, the TCG's focus on movement within a grid allows for easier experimental specification and study of common knowledge rooted in prior expectations based on movements. This makes it an ideal setting to test principles derived from movement kinetics. The expectation violations in the TCG are rooted in deviations from these expected spatial paths, providing a suitable platform for our Surprise Model. (3) Quantifiable Metrics: The game structure allows for precise measurement of behavioral responses, such as the number of steps taken to convey a message and the accuracy of the receiver's interpretation. These metrics provide objective data to validate our computational models and study physiological responses to expectation violations.

(3) In this regard, the possible limits in terms of ecological validity of the results of the present experiment should be better specified. Some suggestions for possible future expansions or modifications of the experimental setting could be added at the end of the paper.

Response: This is an important point, and we recognize the need to address concerns about the ecological validity of our results, which we have done in the discussion section, lines 605-615. Below is the revised paragraph, which we hope answers the reviewer's question.

"While communication through surprise is crucial in the TCG and has broad implications, it is not the exclusive method for novel environments or general communication. The TCG provides a controlled setting for studying non-verbal communication and expectation violations but does not fully capture the complexities of real-world communication, which often involves a mix of verbal and non-verbal

cues and relies on social and emotional contexts. To increase the ecological validity of these results in future research, we could start by translating the game from the non-verbal to the verbal domain. Another modification could involve allowing participants to choose between cooperative and competitive behaviors to explore different social contexts. Finally, we could alter the game design to observe if the intention to communicate can naturally emerge and how surprise plays a role in this process. For instance, participants could learn to communicate while playing the game without explicit instructions to do so.”

(4) The work would benefit from a broader discussion regarding the possible implications of this type of research for application areas (for example, relating to negotiation processes in real interactions).

Response: This was a very helpful point. Here is the revised paragraph that discusses this point in discussion lines 587-603.

“The potential implications of these findings extend across multiple domains, emphasizing the importance of harnessing expectancy violations to improve communication in areas such as negotiation, human-machine interfaces, and AI technologies. In negotiation, utilizing strategic signaling and managing expectations can capture attention and shift dynamics. Beyond negotiation, integrating elements of surprise into human-machine interfaces can make interactions more engaging and intuitive. Unexpected yet relevant system responses can capture user attention and enhance learning and retention, leading to more user-friendly and effective interfaces. Integrating this concept with AI technologies could potentially make AI more human-like and creative. Generative AIs trained on extensive text typically analyze sets of possible words based on contextual information to ensure clarity and coherence by selecting the most probable words. Even though these selection algorithms occasionally introduce some variability through top-k sampling and temperature scaling, they only ensure moderate variability in word selection. Applying the principles of our Surprise model would lead to the selection of greatly unexpected words, thus ensuring even greater variability in the selection process. This adaptation would be a novel approach, potentially diverging from predictable paths to enhance creativity and engagement. Such an application could enhance AI’s capabilities in comedy, storytelling, or generative art, creating more engaging and unique scenarios.”

Reviewer #4:

Remarks to the Author

Response: We thank the reviewer for their time and effort in reviewing the work. We appreciate the assistance in refining the manuscript.

Reviewer #5:

Remarks to the Author

This paper presents two experiments in the Tacit Communication Game framework (where participants combine goal-directed and communicative movement to reach a personal goal while signaling a second location to a partner), and a computational model of this game using a surprise-based framework where signallers construct a movement path which signals goal for the receiver using surprising moves (moves with low prior probability, i.e., moving backwards or away from the eventual goal). The surprise model fits well to participant data, including pupil dilation and EEG data.

The experiment and analysis looks well conducted to me (although I have very little expertise in EEG in particular, so I will defer to other reviewers on that) and the model is well explained and quite intuitive. I found the paper very interesting and exciting and would like to see it published.

Response: We thank the reviewer for their insightful comments and excellent suggestions. Below we tried to answer all the points the reviewer raised.

(1) My only real comment concerns the connection between this work and preceding work from the cognitive science community using similar theoretical notions, experimental methods and related modeling paradigms - I think the paper could be substantially strengthened by drawing out these parallels.

The idea that speakers track the statistical properties of their productions and potentially use those statistical properties to facilitate interpretation is well established, e.g., in the form of the Uniform Information Density hypothesis (works by Roger Levy on this topic would provide a good starting point). Making this parallel explicit early would highlight how these same principles explain “regular” language use (in even more mundane contexts than the humor example you give).

Second, there is literature on iconic communication beyond the tacit communication game that should be acknowledged and can be integrated. This includes papers presenting experiments where participants solve communicative tasks in novel communicative mediums by exploiting the iconic potential of the representational medium they are provided (see e.g., work by Tessa Verhoef on communicating time using spatial movement, work by Bruno Galantucci and Gareth Roberts on other visual communication tasks, Marieke Schouwstra and Yasamin Motamedi on improvised gestural communication, etc etc). While it would be nice to see a broader range of such studies mentioned and, if possible, integrated into your framing, there are some *very* similar movement-based communication paradigms that definitely need to be mentioned, and which fit very nicely within your framework. In particular, Scott-Phillips et al., 2009, “Signalling signalhood”, <https://doi.org/10.1016/j.cognition.2009.08.009>, is clearly relevant in that it involves an experiment in which participants produce flagrantly non-goal-directed movements to signal to their partner; more recent work by Julian Jara-Ettinger (e.g. most obviously <https://escholarship.org/content/qt05z6b1wh/qt05z6b1wh.pdf>, also <https://doi.org/10.1016/j.cognition.2023.105524>, but I think he has numerous relevant papers) use similar tasks and include computational models which are very similar in spirit to those you propose here; it would be fantastic to see these acknowledged, similarities and differences between the frameworks outlined.

Finally, I would like to see how your model integrates with the Rational Speech Act model (developed by Mike Frank and Noah Goodman). In particular, you only really offer an intuitive explanation of why signallers might prefer surprising movements in their messages; as I understand it, under the RSA model this would be understood in terms of the signaller designing a signal which maximizes the posterior probability of a listener interpreting their movement as being directed towards the intended goal, i.e., $p(\text{goal}|\text{signal})$, the sender inverts this to get $p(\text{signal}|\text{goal})$. This sounds very similar in spirit to the Belief-Based Model which you touch on in the supporting materials, but the implementation looks somewhat different. The main thing for me is that the RSA or something potentially similar explains *why* participants would use

surprising moves, from a receiver perspective - it might also allow you to model and capture the receiver success rates from your communication game, which at the moment are left unmodeled but which fall quite naturally into the picture from the RSA perspective.

Response: We appreciate this suggestion, which highlighted a gap in the literature that we had previously overlooked in the submitted version of our manuscript.

Firstly, as recommended, we incorporated the Uniform Information Density hypothesis³ and the Rational Speech Act (RSA)² model to emphasize the primary drivers of natural verbal communication. Both theories highlight the importance of tracking the statistical properties of speech or text and selecting the most predictable outcomes from the Receiver's perspective, given that both speakers and listeners are rational agents who aim to maximize communicative success. We discuss these theories together, illustrating how naturalistic verbal communication is driven by predictability (lines 27-35). Here is the revised paragraph that implements these changes.

“Effective communication relies on the ability of the Sender to design messages that clearly convey intentions, and the ability of the Receiver to accurately interpret these messages. In verbal communication, this is achieved as speakers and listeners track the statistical properties of language to optimize communication, often selecting the most predictable outcomes to facilitate understanding¹. The Rational Speech Act model, for instance, posits that speakers optimize utterances to minimize listener's surprise while maximizing the likelihood of accurate interpretation by the listener². Similarly, the Information Density Hypothesis suggests that speakers manipulate syntactic structures to maintain a consistent flow of information, by aligning their linguistic choices with the statistical regularities of speech³.”

Following this, we integrated the literature on iconic communication, including the suggested papers and additional relevant studies⁴⁻⁶ in lines 37-47. Here is the corresponding paragraph:

“However, this predictability, seen in natural language processing, is not always maintained in communication. In many instances, communication takes on unconventional forms to achieve a specific goal. For example, in signaling behavior where the Sender needs to strategically direct the attention of the Receiver, it has been demonstrated that inefficiencies can signal particular intentions⁴, objects, even those not intrinsically communicative like chairs or ropes, can convey social meanings through their contextual arrangement⁵, and non-verbal cues can embody complex concepts like time⁶. These examples, while diverse in communicative strategies, highlight a common theme: the strategic deviation from expected communicative patterns to achieve greater clarity and intentionality in communication. These deviations are particularly effective because they contrast sharply with anticipated behaviors, thereby capturing attention and signaling intentions more distinctly.”

Lastly, we found it very interesting to compare RSA to the surprise model. As we understand, the RSA model explains that speakers (or senders in the case of TCG) choose signals (or movements for TCG) that optimize the receiver's understanding, considering their shared knowledge and expectations. The surprising movements in our model can be considered a strategic choice to enhance the informativeness and clarity of the signal by deviating from this shared knowledge.

The difference between RSA and the surprise model is that RSA tries to minimize the Receiver's surprise, while the Surprise model maximizes it. In RSA, Speakers aim to decide signals (utterances) that maximize the likelihood that the listener will correctly infer their intended meaning. This involves minimizing the listener's surprise by providing informative and relevant utterances based on shared knowledge and expectations.

We recognize the importance of discussing the RSA model in more detail. However, due to the extensive new implementations and the constraints of the word count, we have only briefly mentioned the RSA model as mentioned before in the context of communication in naturalistic settings. We hope this sufficiently addresses the reviewer's suggestions.

Minor Suggestions

(1) Figure 1 could be better explained in the caption, I had to do quite a lot of work to infer what was going on. In particular, in figure A, please provide a legend explaining what the circle and cross symbols mean; in 1d I really could not see how that message profile was decomposed into the sequence of movements given, perhaps because I don't understand the implicit facing of the circle character icon.

Response: This is a good point, and we thank the reviewer for raising it. Firstly, what was referred to as Figure 1 in the original manuscript is now Figure 2 in the revised manuscript. We have now included explanations in the captions to address these points. Specifically, in Figure 2a (lines 138-141), we have defined the meanings of the different colored circles and crosses. The orange circle and cross represent the starting position and goal location for the Sender, respectively, while the blue circle and cross indicate the starting position and goal location for the Receiver. In Figure 2d, we have clarified the movement directions (lines 149-153). The orange circle marks the starting location. The direction of movement is always defined relative to the previous movement. The initial movement originating from the starting location does not have a predefined direction. However, subsequent movements are oriented based on the preceding movement. If a movement continues in the same path as the previous one, it is considered 'forward.' Other movement directions (e.g., left, right) are defined relative to this initial forward movement. Figure 4b in this response letter illustrates how directions (and therefore probabilities) shift with moving around on the grid.

(2) Table 1, the scientific notation for PEP values is not very user-friendly and this level of precision is probably not required!

Response: We thank the reviewer for pointing this out. We have now rounded the PEP values to fewer digits. We hope that the table becomes more readable after this change (see page 12, table 1 for changes).

(3) Statistical analysis of pupil response - I wasn't sure why binning of the surprise level was required, rather than treating it continuously as for the EEG data, but in any case I think you need a by-participant random slope for surprise level in your model here (equation 1 on page 21).

Response: We thank the reviewer for this suggestion. Reviewer #2 raised the same question, and we have provided a detailed explanation in our response on pages 15-16. In summary, we decided to use surprise as the continuous variable and updated the text in the manuscript accordingly. Additionally, we agree with the reviewer's suggestion and have included a by-participant random slope for the surprise level in our model.

References mentioned in this letter:

1. Ryskin, R. & Nieuwland, M. S. Prediction during language comprehension: what is next? *Trends Cogn. Sci.* **27**, 1032–1052 (2023).
2. Goodman, N. D. & Frank, M. C. Pragmatic Language Interpretation as Probabilistic Inference. *Trends Cogn. Sci.* **20**, 818–829 (2016).
3. Levy, R. & Jaeger, T. Speakers optimize information density through syntactic reduction. *Adv. Neural Inf. Process. Syst.* 849–856 (2006) doi:10.7551/mitpress/7503.003.0111.
4. Royka, A., Chen, A., Aboody, R., Huanca, T. & Jara-Ettinger, J. People infer communicative action through an expectation for efficient communication. *Nat.*

- Commun.* **13**, 4160 (2022).
5. Lopez-Brau, M. & Jara-Ettinger, J. People can use the placement of objects to infer communicative goals. *Cognition* **239**, 105524 (2023).
 6. Verhoef, T., Walker, E. & Marghetis, T. Interaction dynamics affect the emergence of compositional structure in cultural transmission of space-time mappings. *Proceedings of the Annual Meeting of the Cognitive Science Society* **44**, (2022).
 7. Shain, C., Meister, C., Pimentel, T., Cotterell, R. & Levy, R. Large-scale evidence for logarithmic effects of word predictability on reading time. *Proc. Natl. Acad. Sci. U. S. A.* **121**, e2307876121 (2024).
 8. Shannon, C. E. A mathematical theory of communication. *SIGMOBILE Mob. Comput. Commun. Rev.* **5**, 3–55 (2001).
 9. Rohde, H. & Rubio-Fernandez, P. Color interpretation is guided by informativity expectations, not by world knowledge about colors. *J. Mem. Lang.* **127**, 104371 (2022).
 10. Shaw, J. Philosophy of humor. *Philos. Compass* **5**, 112–126 (2010).
 11. Kurumada, C., Brown, M., Bibyk, S., Pontillo, D. F. & Tanenhaus, M. K. Is it or isn't it: listeners make rapid use of prosody to infer speaker meanings. *Cognition* **133**, 335–342 (2014).
 12. Stephens, R. C. & Ward, J. J. Motion In A Circle. in *Applied Mechanics* (eds. Stephens, R. C. & Ward, J. J.) 27–39 (Macmillan Education UK, London, 1972). doi:10.1007/978-1-349-00870-4_3.
 13. Baker, C. L., Jara-Ettinger, J., Saxe, R. & Tenenbaum, J. B. Rational quantitative attribution of beliefs, desires and percepts in human mentalizing. *Nature Human Behaviour* **1**, 1–10 (2017).

14. Yuan, L. Emergence of Theory of Mind Collaboration in Multiagent Systems. 1–11 (2019).
15. De Weerd, H., Verbrugge, R. & Verheij, B. How much does it help to know what she knows you know? An agent-based simulation study. *Artif. Intell.* **199-200**, 67–92 (2013).
16. de Weerd, H., Verbrugge, R. & Verheij, B. Higher-order theory of mind in the Tacit Communication Game. *Biologically Inspired Cognitive Architectures* **11**, 10–21 (2015).
17. Preuschoff, K., 't Hart, B. M. & Einhäuser, W. Pupil Dilation Signals Surprise: Evidence for Noradrenaline's Role in Decision Making. *Front. Neurosci.* **5**, 115 (2011).
18. Luke, S. G. Evaluating significance in linear mixed-effects models in R. *Behav. Res. Methods* **49**, 1494–1502 (2017).
19. Nachev, P., Kennard, C. & Husain, M. Functional role of the supplementary and pre-supplementary motor areas. *Nat. Rev. Neurosci.* **9**, 856–869 (2008).
20. Kolling, N., Behrens, T., Wittmann, M. K. & Rushworth, M. Multiple signals in anterior cingulate cortex. *Curr. Opin. Neurobiol.* **37**, 36–43 (2016).
21. Kennerley, S. W., Walton, M. E., Behrens, T. E. J., Buckley, M. J. & Rushworth, M. F. S. Optimal decision making and the anterior cingulate cortex. *Nat. Neurosci.* **9**, 940–947 (2006).
22. Rushworth, M. F. S., Kolling, N., Sallet, J. & Mars, R. B. Valuation and decision-making in frontal cortex: one or many serial or parallel systems? *Curr. Opin. Neurobiol.* **22**, 946–955 (2012).
23. Rushworth, M. F. S., Walton, M. E., Kennerley, S. W. & Bannerman, D. M. Action sets and decisions in the medial frontal cortex. *Trends Cogn. Sci.* **8**, 410–417 (2004).
24. Apps, M. A. J., Rushworth, M. F. S. & Chang, S. W. C. The Anterior Cingulate Gyrus

- and Social Cognition: Tracking the Motivation of Others. *Neuron* **90**, 692–707 (2016).
25. Zhang, L. & Gläscher, J. A brain network supporting social influences in human decision-making. *Sci Adv* **6**, eabb4159 (2020).
 26. Holroyd, C. B. & Coles, M. G. H. The neural basis of human error processing: Reinforcement learning, dopamine, and the error-related negativity. *Psychol. Rev.* **109**, 679–709 (2002).
 27. Ruiter, J. P. D. *et al.* Exploring the cognitive infrastructure of communication. *Interaction StudiesInteraction Studies. Social Behaviour and Communication in Biological and Artificial Systems* **11**, 51–77 (2010).
 28. Haggard, P. *et al.* On the origin of intentions. *Sensorimotor Foundations of Higher Cognition* 601–618 (2012) doi:10.1093/acprof:oso/9780199231447.003.0026.
 29. Blokpoel, M. *et al.* Recipient design in human communication: simple heuristics or perspective taking? *Front. Hum. Neurosci.* **6**, 253 (2012).
 30. Willems, R. M., Benn, Y., Hagoort, P., Toni, I. & Varley, R. Communicating without a functioning language system: Implications for the role of language in mentalizing. *Neuropsychologia* **49**, 3130–3135 (2011).

Dear reviewers,

Thank you so much for taking the time to review our manuscript and the response letter we provided. We appreciate your thoughtful feedback, which has been invaluable in refining our work. Below, we address the remaining comments.

Reviewer #1:

Remarks to the Author

We thank the authors for the careful revisions of the original manuscript, which we believed have greatly improved their paper. We are very satisfied with their response to our queries and look forward to seeing this work published in Nature Communications.

Response: Thank you so much for your kind words and thoughtful feedback during the review process. We are glad to hear that you approve our revisions, and we are excited about the chance to share this work with the community!

Reviewer #2:

Remarks to the Author

The authors have addressed the majority of my concerns, and I think the manuscript is much improved.

(1) I remain somewhat less convinced by the EEG analysis. What the authors refer to as N300 and N400 don't really align with those components based on the amplitude patterns and topography. N400s for instance are typically centro-parietally distributed. Also, the peaks appear to be more positive for high surprise events which is the reverse of the expected pattern. I think it would be somewhat more reasonable to interpret this as a prolonged P300. In any case, I think it would be sufficient to temper the conclusions about the EEG results in the main text.

Response: We thank the reviewer for thoughtful and constructive feedback regarding the EEG analysis. We acknowledge the point that the observed EEG effects do not align neatly with canonical N300 or N400 components, particularly in terms of amplitude patterns and topography. We also appreciate the suggestion to temper the conclusions regarding the EEG results.

To address these concerns, we have taken the following steps:

- 1) We do not explicitly refer to the observed EEG clusters as N300, N400, or P300 components in the text. Instead, our interpretation focuses on source-level findings, linking the observed time-space clusters to neural activity in the SMA and ACC regions.
- 2) Considering the reviewer's suggestion, we have rewritten the relevant paragraph in the discussion (lines 480–496) to tone down the interpretation of the EEG results. Specifically, we now emphasize the exploratory nature of these findings and highlight the need for further studies to confirm the functional specificity of the observed effects. Here is the revised paragraph from the discussion:

“Sensor-level analysis of the neural encoding of surprise in the EEG data revealed two prominent time-space clusters over frontal and fronto-central electrodes. Source localization analysis identified a dipole, the potential neuronal origin of these clusters, in the region encompassing both the Supplementary Motor Area (SMA) and the dorsal anterior cingulate cortex (dACC). The first cluster, appearing around 400 ms after the onset of the communication step, may reflect the Receiver mentally simulating the Sender's intended motor actions while observing the message, potentially involving SMA activity, which is typically engaged in motor planning⁴⁹. The second cluster, observed later (800ms after the onset), may be associated with the detection of deviations from priors, involving the (dorsal) anterior cingulate cortex (ACC), a region known for error detection and the integration of diverse information to guide decisions aligned with goals and reward representations^{15,16}. The ACC is also thought to play a crucial role in linking actions to outcomes and adjusting perceptions based on new evidence^{50,51}. Recent studies suggest its involvement in discerning others' intentions and detecting unexpected outcomes^{52,53}. Our findings align with this perspective, and we interpret the second cluster as potentially reflecting ACC activity related to the Receiver's processing of surprise as intentionally designed by the Sender²². However, we recognize that these interpretations remain exploratory, and further studies are needed to confirm the functional specificity of these findings.”

(2) As a more general point, there are a few compelling analyses and figures that appear in the response letter but not the main text. I would recommend adding at least a footnote to briefly provide the results of these analyses since many readers may have the same questions that I had as a reviewer (e.g., participants' strategies in the game, the role of step number imbalances).

Response: We agree that some of the analyses and figures presented in the previous response letter address potential questions that readers might also have. To ensure accessibility to this information, we will include these analyses and figures in the supplementary materials and provide clear references to them in the main text.

(3) Finally, I wasn't able to look at the new code because the repo appears to have been made private now.

Response: We apologize for the inconvenience you experienced. It was always our intention to make the code available throughout the review process to ensure transparency and reproducibility. However, we acknowledge that there may have been a glitch in the accessibility settings. We have now double-checked and confirmed that the repository is fully accessible.

Reviewer #4:

Remarks to the Author

Response: We thank the reviewer for their time and effort in reviewing the work. We appreciate the assistance in this process.

Reviewer #5:

Remarks to the Author

I liked the first draft of this paper, and the authors have responded appropriately to my suggestions, so I am happy to see it published. My one remaining quibble is that I think they overstate the mismatch between their approach and the conventional one from linguistics, e.g. on lines 491-501 where they say "The key insight of this study is that a framework of expectation violation can serve as a mechanism for signaling goal orientation in human communication. ... This approach contrasts with traditional communication models that rely on predictability and shared linguistic cues." I think a real strength of their experiment is that it provides a beautifully controlled environment for studying how violation of expectation is communicative, but this has long been observed for natural language too, where the prior expectations are based on knowledge of a conventional grammar and assumptions about speaker efficiency. For example, in the classic work on implicature by Laurence Horn (e.g. reviewed in <https://doi.org/10.1002/9780470756959.ch1>), it is observed that being more verbose than

necessary can convey additional information, so if i say "I got the car to start", that conveys additional information on top of "I started the car" precisely *because* it's unnecessarily verbose and violates the expectation you should have that I shouldn't be aimlessly verbose. But that tweak notwithstanding, I think this is an impressive and exciting paper that I'd like to see published.

Response: We thank the reviewer for the thoughtful observation and for highlighting the need to better align our discussion with existing results from natural language communication. We agree that the use of expectation violations in communication has been observed in natural language contexts, as exemplified by implicature and prosody.

To address this, we added a following paragraph to the discussion (lines 411-421):

“While predictability is the primary mechanism in classic communication models, there are situations where expectancy violations can also play a crucial role in effective communication. For example, classic work on implicature shows that being unnecessarily verbose can convey additional meaning because the verbosity violates the expectation of conciseness³⁰. Similarly, in prosody, unexpected variations in pitch, tone, or rhythm can enhance the emotional impact and engagement of verbal communication³¹. More recent research has explored the adaptive benefits of expectancy violations. Repeated exposure to syntactic anomalies, such as garden path sentences, helps individuals adapt and process language more effectively over time³². Moreover, strategically introducing unexpected lexical or syntactic complexity can also enhance the persuasive impact of messages, demonstrating how deliberate expectancy violations improve communication effectiveness, even in contexts requiring high levels of clarity and engagement³³.”

We hope this revision addresses the reviewer's concern and better situates our findings within the broader context of communication theory. Thank you again for this valuable feedback.